# LESS IS MORE: FEWER INTERPRETABLE REGION VIA SUBMODULAR SUBSET SELECTION

**Ruoyu Chen**[1,2], **Hua Zhang**[1,2,*], **Siyuan Liang**[3], **Jingzhi Li**[1,2], **Xiaochun Cao**[4]

[1]Institute of Information Engineering, Chinese Academy of Sciences, Beijing 100093, China
[2]School of Cyber Security, University of Chinese Academy of Sciences, Beijing 100049, China
`{chenruoyu,zhanghua,lijingzhi}@iie.ac.cn`
[3]School of Computing, National University of Singapore, 119077, Singapore
`pandaliang521@gmail.com`
[4]School of Cyber Science and Technology, Shenzhen Campus of Sun Yat-sen University,
Shenzhen 518107, China
`caoxiaochun@mail.sysu.edu.cn`

## ABSTRACT

Image attribution algorithms aim to identify important regions that are highly relevant to model decisions. Although existing attribution solutions can effectively assign importance to target elements, they still face the following challenges: 1) existing attribution methods generate inaccurate small regions thus misleading the direction of correct attribution, and 2) the model cannot produce good attribution results for samples with wrong predictions. To address the above challenges, this paper re-models the above image attribution problem as a submodular subset selection problem, aiming to enhance model interpretability using fewer regions. To address the lack of attention to local regions, we construct a novel submodular function to discover more accurate small interpretation regions. To enhance the attribution effect for all samples, we also impose four different constraints on the selection of sub-regions, i.e., confidence, effectiveness, consistency, and collaboration scores, to assess the importance of various subsets. Moreover, our theoretical analysis substantiates that the proposed function is in fact submodular. Extensive experiments show that the proposed method outperforms SOTA methods on two face datasets (Celeb-A and VGG-Face2) and one fine-grained dataset (CUB-200-2011). For correctly predicted samples, the proposed method improves the Deletion and Insertion scores with an average of 4.9% and 2.5% gain relative to HSIC-Attribution. For incorrectly predicted samples, our method achieves gains of 81.0% and 18.4% compared to the HSIC-Attribution algorithm in the average highest confidence and Insertion score respectively. The code is released at https://github.com/RuoyuChen10/SMDL-Attribution.

## 1 INTRODUCTION

Building transparent and explainable artificial intelligence (XAI) models is crucial for humans to reasonably and effectively exploit artificial intelligence (Dwivedi et al., 2023; Ya et al., 2024; Li et al., 2021b; Tu et al., 2023; Liang et al., 2022a;b; 2023b). Solving the interpretability and security problems of AI has become an urgent challenge in the current field of AI research (Arrieta et al., 2020; Chen et al., 2023; Zhao et al., 2023; Liang et al., 2023a; 2024; Li et al., 2023). Image attribution algorithms are typical interpretable methods, which produce saliency maps that explain which image regions are more important to model decisions. It provides a deeper understanding of the operating mechanisms of deep models, aids in identifying sources of prediction errors, and contributes to model improvement. Image attribution algorithms encompass two primary categories: white-box methods (Chattopadhay et al., 2018; Wang et al., 2020) chiefly attribute on the basis of the model's internal characteristics or decision gradients, and black-box methods (Petsiuk et al., 2018; Novello et al., 2022), which primarily analyze the significance of input regions via external perturbations.

Although attribution algorithms have been well studied, they still face two following limitations: 1) Some attribution regions are not small and dense enough which may interfere with the optimization

---

*Corresponding Author

Figure 1: For correctly predicted samples, our method can find fewer regions that make the model predictions more confident. For incorrectly predicted samples, our method can find the reasons that caused the model to predict incorrectly.

orientation. Most advanced attribution algorithms focus on understanding how the global image contributes to the prediction results of the deep model while ignoring the impact of local regions on the results. 2) It is difficult to effectively attribute the latent cause of the error to samples with incorrect predictions. For incorrectly predicted samples, although the attribution algorithm can assign the correct class, it does not limit the response to the incorrect class, so that the attributed region may still have a high response to the incorrect class.

To solve the above issues, we propose a novel explainable method that reformulates the attribution problem as a submodular subset selection problem. We hypothesize that local regions can achieve better interpretability, and hope to achieve higher interpretability with fewer regions. We first divide the image into multiple sub-regions and achieve attribution by selecting a fixed number of sub-regions to achieve the best interpretability. *To alleviate the insufficient dense of the attribution region*, we employ regional search to continuously expand the sub-region set. Furthermore, we introduce a new attribution mechanism to excavate what regions promote interpretability from four aspects, i.e., the prediction confidence of the model, the effectiveness of regional semantics, the consistency of semantics, and the collective effect of the region. Based on these useful clues, we design a submodular function to evaluate the significance of various subsets, which can *limit the search for regions with wrong class responses*. As shown in Fig. 1, we find that for correctly predicted samples, the proposed method can obtain higher prediction confidence with only a few regions as input than with all regions as input, and for incorrectly predicted samples, our method has the ability to find the reasons (e.g., the dark background region) that caused its prediction error.

We validate the effectiveness of our method on two face datasets, i.e., Celeb-A (Liu et al., 2015), VGG-Face2 (Cao et al., 2018), and a fine-grained dataset CUB-200-2011 (Welinder et al., 2010). For correctly predicted samples, compared with the SOTA method HSIC-Attribution (Novello et al., 2022), our method improves the Deletion scores by 8.4%, 1.0%, and 5.3%, and the Insertion scores by 1.1%, 0.2%, and 6.1%, respectively. For incorrectly predicted samples, our method excels in identifying the reasons behind the model's decision-making errors. In particular, compared with the SOTA method HSIC-Attribution, the average highest confidence is increased by 81.0%, and the Insertion score is increased by 18.4% on the CUB-200-2011 dataset. Furthermore, our analysis at the theoretical level confirms that the proposed function is indeed submodular. Please see Appendix 5 for a list of important notations employed in this article.

Our contributions can be summarized as follows:

- We reformulate the attribution problem as a submodular subset selection problem that achieves higher interpretability with fewer dense regions.

- A novel submodular mechanism is constructed to evaluate the significance of various subsets from four different aspects, i.e., the confidence, effectiveness, consistency, and collaboration scores. It not only improves the dense of attribution regions for existing attribution algorithms but also better discovers the causes of image prediction errors.

- The proposed method has good versatility which can improve the interpretability of both on the face and fine-grained datasets. Experimental results show that it can achieve better attribution results for the correctly predicted samples and effectively discover the causes of model prediction errors for incorrectly predicted samples.

## 2 RELATED WORK

**White-Box Attribution Method** Image attribution algorithms are designed to ascertain the importance of different input regions within an image with respect to the decision-making process of the model. White-box attribution algorithms primarily rely on computing gradients with respect to

the model's decisions. Early research focused on assessing the importance of individual input pixels in image-based model decision-making (Baehrens et al., 2010; Simonyan et al., 2014). Some methods propose gradient integration to address the vanishing gradient problem encountered in earlier approaches (Sundararajan et al., 2017; Smilkov et al., 2017). Recent popular methods often focus on feature-level activations in neural networks, such as CAM (Zhou et al., 2016), Grad-CAM (Selvaraju et al., 2020), Grad-CAM++ (Chattopadhay et al., 2018), and Score-CAM (Wang et al., 2020). However, these methods all rely on the selection of network layers in the model, which has a greater impact on the quality of interpretation (Novello et al., 2022).

**Black-Box Attribution Method** Black-box attribution methods assume that the internal structure of the model is agnostic. LIME (Ribeiro et al., 2016) locally approximates the predictions of a black-box model with a linear model, by only slightly perturbing the input. RISE (Petsiuk et al., 2018) perturbs the model by inputting multiple random masks and weighting the masks to get the final saliency map. HSIC-Attribution method (Novello et al., 2022) measures the dependence between the input image and the model output through Hilbert Schmidt Independence Criterion (HSIC) based on RISE. Petsiuk et al. (2021) propose D-RISE to explain the object detection model's decision.

**Submodular Optimization** Submodular optimization (Fujishige, 2005) has been successfully studied in multiple application scenarios (Wei et al., 2014; Yang et al., 2019; Kothawade et al., 2022; Joseph et al., 2019), a small amount of work also uses its theory to do research related to model interpretability. Elenberg et al. (2017) frame the interpretability of black-box classifiers as a combinatorial maximization problem, it achieves similar results to LIME and is more efficient. Chen et al. (2018) introduce instance-wise feature selection to explain the deep model. Pervez et al. (2023) proposed a simple subset sampling alternative based on conditional Poisson sampling, which they applied to interpret both image and text recognition tasks. However, these methods only retained the selected important pixels and observed the recognition accuracy (Chen et al., 2018). In this article, we elucidate our model based on submodular subset selection theory and attain state-of-the-art results, as assessed using standard metrics for attribution algorithm evaluation. Our method also effectively highlights factors leading to incorrect model decisions.

## 3 PRELIMINARIES

In this section, we first establish some definitions. Considering a finite set $V$, given a set function $\mathcal{F} : 2^V \to \mathbb{R}$ that maps any subset $S \subseteq V$ to a real value. When $\mathcal{F}$ is a submodular function, its definition is as follows:

**Definition 3.1** (Submodular function (Edmonds, 1970)). For any set $S_a \subseteq S_b \subseteq V$. Given an element $\alpha$, where $\alpha = V \setminus S_b$. The set function $\mathcal{F}$ is a submodular function when it satisfies monotonically non-decreasing ($\mathcal{F}(S_b \cup \{\alpha\}) - \mathcal{F}(S_b) \geq 0$) and:

$$\mathcal{F}(S_a \cup \{\alpha\}) - \mathcal{F}(S_a) \geq \mathcal{F}(S_b \cup \{\alpha\}) - \mathcal{F}(S_b). \tag{1}$$

**Problem formulation** We divide an image $\mathbf{I}$ into a finite number of sub-regions, denoted as $V = \left\{ \mathbf{I}_1^M, \mathbf{I}_2^M, \cdots, \mathbf{I}_m^M \right\}$, where $M$ indicates a sub-region $\mathbf{I}^M$ formed by masking part of image $\mathbf{I}$. Giving a monotonically non-decreasing submodular function $\mathcal{F} : 2^V \to \mathbb{R}$, the image recognition attribution problem can be viewed as maximizing the value $\mathcal{F}(S)$ with limited regions. Mathematically, the goal is to select a set $S$ consisting of a limited number $k$ of sub-regions in the set $V$ that maximize the submodular function $\mathcal{F}$:

$$\max_{S \subseteq V, |S| \leq k} \mathcal{F}(S), \tag{2}$$

we can transform the image attribution problem into a subset selection problem, where the submodular function $\mathcal{F}$ relates design to interpretability.

## 4 PROPOSED METHOD

In this section, we propose a novel method for image attribution based on submodular subset selection theory. In Sec. 4.1 we introduce how to perform sub-region division, in Sec. 4.2 we introduce our designed submodular function, and in Sec. 4.3 we introduce attribution algorithm based on greedy search. Fig. 2 shows the overall framework of our approach.

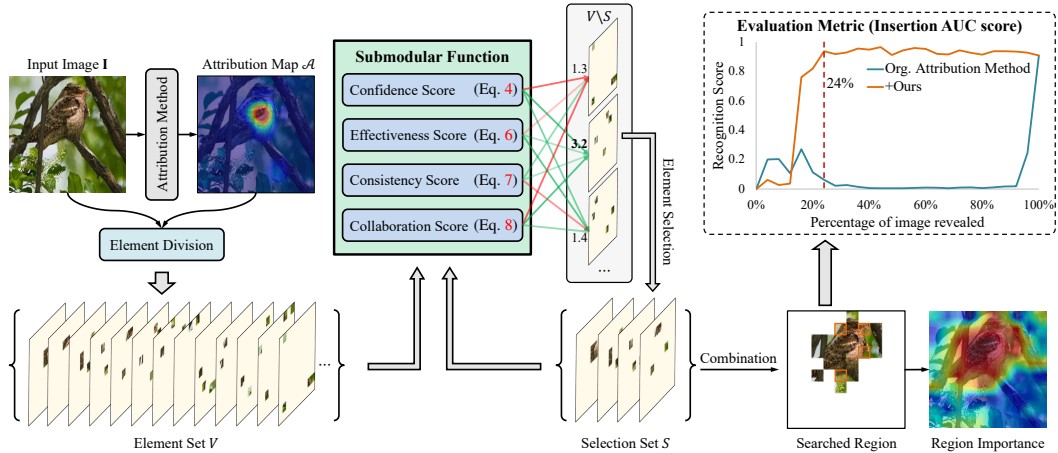

Figure 2: The framework of the proposed method.

## 4.1 SUB-REGION DIVISION

To obtain the interpretable region in an image, we partition the image $\mathbf{I} \in \mathbb{R}^{w \times h \times 3}$ into $m$ sub-regions $\mathbf{I}^M$, where $M$ indicates a sub-region $\mathbf{I}^M$ formed by masking part of image $\mathbf{I}$. Traditional methods (Noroozi & Favaro, 2016; Redmon & Farhadi, 2018) typically divide images into regular patch areas, neglecting the semantic information inherent in different regions. In contrast, our method employs a sub-region division strategy that is informed and guided by an a priori saliency map. In detail, we initially divide the image into $N \times N$ patch regions. Subsequently, an existing image attribution algorithm is applied to compute the saliency map $\mathcal{A} \in \mathbb{R}^{w \times h}$ for a corresponding class of $\mathbf{I}$. Following this, we resize $\mathcal{A}$ to a $N \times N$ dimension, where its values denote the importance of each patch. Based on the determined importance of each patch, $d$ patches are sequentially allocated to each sub-region $\mathbf{I}^M$, while the remaining patch regions are masked with $\mathbf{0}$, where $d = N \times N / m$. This process finally forms the element set $V = \left\{ \mathbf{I}_1^M, \mathbf{I}_2^M, \cdots, \mathbf{I}_m^M \right\}$, satisfying the condition $\mathbf{I} = \sum_{i=1}^m \mathbf{I}_i^M$. The detailed calculation process is outlined in Algorithm 1.

## 4.2 SUBMODULAR FUNCTION DESIGN

**Confidence Score** We adopt a model trained by evidential deep learning (EDL) to quantify the uncertainty of a sample (Sensoy et al., 2018), which is developed to overcome the limitation of softmax-based networks in the open-set environment (Bao et al., 2021). We denote the network with prediction uncertainty by $F_u(\cdot)$. For $K$-class classification, assume that the predicted class probability values follow a Dirichlet distribution. When given a sample $\mathbf{x}$, the network is optimized by the following loss function:

$$\mathcal{L}_{EDL} = \sum_{k_c=1}^{K} \mathbf{y}_{k_c} \left( \log S_{\text{Dir}} - \log \left( \mathbf{e}_{k_c} + 1 \right) \right),\tag{3}$$

where $\mathbf{y}$ is the one-hot label of the sample $\mathbf{x}$, and $\mathbf{e} \in \mathbb{R}^K$ is the output and the learned evidence of the network, denoted as $\mathbf{e} = \exp \left( F_u(\mathbf{x}) \right)$. $S_{\text{Dir}} = \sum_{k_c=1}^{K} \left( \mathbf{e}_{k_c} + 1 \right)$ is referred to as the Dirichlet strength. In the inference, the predictive uncertainty can be calculated as $u = K / S_{\text{Dir}}$, where $u \in [0, 1]$. Thus, the confidence score of the sample $\mathbf{x}$ predicted by the network can be expressed as:

$$s_{\text{conf.}}(\mathbf{x}) = 1 - u = 1 - \frac{K}{\sum_{k_c=1}^{K} \left( \mathbf{e}_{k_c} + 1 \right)}.\tag{4}$$

By incorporating the $s_{\text{conf.}}$, we can ensure that the selected regions align closely with the In-Distribution (InD). This score acts as a reliable metric to distinguish regions from out-of-distribution, ensuring alignment with the InD.

**Effectiveness Score** We expect to maximize the response of valuable information with fewer regions since some image regions have the same semantic representation. Given an element $\alpha$, and a

sub-set $S$, we measure the distance between the element $\alpha$ and all elements in the set, and calculate the smallest distance, as the effectiveness score of the judgment element $\alpha$ for the sub-set $S$:

$$s_e\left(\alpha \mid S\right) = \min_{s_i \in S} \text{dist}\left(F\left(\alpha\right), F\left(s_i\right)\right), \tag{5}$$

where $\text{dist}(\cdot, \cdot)$ denotes the equation to calculate the distance between two elements. Traditional distance measurement methods (Wang et al., 2018; Deng et al., 2019) are tailored to maximize the decision margins between classes during model training, involving operations like feature scaling and increasing angle margins. In contrast, our method focuses solely on calculating the relative distance between features, for which we utilize the general cosine distance. $F(\cdot)$ denotes a pre-trained feature extractor. To calculate the element effectiveness score of a set, we can compute the sum of the effectiveness scores for each element:

$$s_{\text{eff.}}\left(S\right) = \sum_{s_i \in S} \min_{s_j \in S, s_i \neq s_j} \text{dist}\left(F\left(s_i\right), F\left(s_j\right)\right). \tag{6}$$

By incorporating the $s_{\text{eff.}}$, we aim to limit the selection of regions with similar semantic representations, thereby increasing the diversity and improving the overall quality of region selection.

**Consistency Score** As the scores mentioned previously may be maximized by regions that aren't target-dependent, we aim to make the representation of the identified image region consistent with the original semantics. Given a target semantic feature, $\boldsymbol{f}_s$, we make the semantic features of the searched image region close to the target semantic features. We introduce the consistency score:

$$s_{\text{cons.}}\left(S, \boldsymbol{f}_s\right) = \frac{F\left(\sum_{\mathbf{I}^M \in S} \mathbf{I}^M\right) \cdot \boldsymbol{f}_s}{\left\|F\left(\sum_{\mathbf{I}^M \in S} \mathbf{I}^M\right)\right\| \|\boldsymbol{f}_s\|}, \tag{7}$$

where $F(\cdot)$ denotes a pre-trained feature extractor. The target semantic feature $\boldsymbol{f}_s$, can either adopt the features computed from the original image using the pre-trained feature extractor, expressed as $\boldsymbol{f}_s = F\left(\mathbf{I}\right)$, or directly implement the fully connected layer of the classifier for a specified class. By incorporating the $s_{\text{cons.}}$, our method targets regions that reinforce the desired semantic response. This approach ensures a precise selection that aligns closely with our specific semantic goals.

**Collaboration Score** Some individual elements may lack significant individual effects in isolation, but when placed within the context of a group or system, they exhibit an indispensable collective effect. Therefore, we introduce the collaboration score, which is defined as:

$$s_{\text{colla.}}\left(S, \mathbf{I}, \boldsymbol{f}_s\right) = 1 - \frac{F\left(\mathbf{I} - \sum_{\mathbf{I}^M \in S} \mathbf{I}^M\right) \cdot \boldsymbol{f}_s}{\left\|F\left(\mathbf{I} - \sum_{\mathbf{I}^M \in S} \mathbf{I}^M\right)\right\| \|\boldsymbol{f}_s\|}, \tag{8}$$

where $F(\cdot)$ denotes a pre-trained feature extractor, $\boldsymbol{f}_s$ is the target semantic feature. By introducing the collaboration score, we can judge the collective effect of the element. By incorporating the $s_{\text{colla.}}$, our method pinpoints regions whose exclusion markedly affects the model's predictive confidence. This effect underscores the pivotal role of these regions, indicative of a significant collective impact. Such a metric is particularly valuable in the initial stages of the search, highlighting regions essential for sustaining the model's accuracy and reliability.

**Submodular Function** We construct our objective function for selecting elements through a combination of the above scores, $\mathcal{F}(S)$, as follows:

$$\mathcal{F}(S) = \lambda_1 s_{\text{conf.}}\left(\sum_{\mathbf{I}^M \in S} \mathbf{I}^M\right) + \lambda_2 s_{\text{eff.}}\left(S\right) + \lambda_3 s_{\text{cons.}}\left(S, \boldsymbol{f}_s\right) + \lambda_4 s_{\text{colla.}}\left(S, \mathbf{I}, \boldsymbol{f}_s\right), \tag{9}$$

where $\lambda_1, \lambda_2, \lambda_3$, and $\lambda_4$ represent the weighting factors used to balance each score. To simplify parameter adjustment, all weighting coefficients are set to 1 by default.

**Lemma 1.** *Consider two sub-sets $S_a$ and $S_b$ in set $V$, where $S_a \subseteq S_b \subseteq V$. Given an element $\alpha$, where $\alpha \subseteq V \setminus S_b$. Assuming that $\alpha$ is contributing to model interpretation, then, the function $\mathcal{F}(\cdot)$ in Eq. 9 is a submodular function and satisfies Eq. 1.*

*Proof.* Please see Appendix B.1 for the proof. □

**Lemma 2.** *Consider a subset $S$, given an element $\alpha$, assuming that $\alpha$ is contributing to model interpretation. The function $\mathcal{F}(\cdot)$ of Eq. 9 is monotonically non-decreasing.*

*Proof.* Please see Appendix B.2 for the proof. □

### 4.3 GREEDY SEARCH ALGORITHM

---

**Algorithm 1:** A greedy search based algorithm for interpretable region discovery

---

**Input:** Image $\mathbf{I} \in \mathbb{R}^{w \times h \times 3}$, number of divided patches $N \times N$, a prior saliency map $\mathcal{A} \in \mathbb{R}^{w \times h \times 3}$ of $\mathbf{I}$, number of image division sub-regions $m$, maximum number of constituent regions $k$.

**Output:** A set $S \subseteq V$, where $|S| \leq k$.

```
1   V ← ∅ ;                                        /* Initiate the operation of sub-region division */
2   A ← resize (A, newRows = N, newCols = N) ;
3   d = N × N/m ;
4   for l = 1 to m do
5       I_l^M = I ;
6       for i = 1 to N do
7           for j = 1 to N do
8               I_r = rank (A, i, j) ;                   /* Index of A_{i,j}, ordered descendingly */
9               if I_r < (d − 1) × l and I_r > d × l then
10                  I_l^M [(I_r − 1) × h/N + 1 : I_r × h/N, (I_r − 1) × w/N + 1 : I_r × w/N] = 0 ;   /* Mask
                        non-relevant patch region, fill it with 0 */
11              end
12          end
13      V ← V ∪ {I_l^M} ;
14  end
15  S ← ∅ ;                                         /* Initiate the operation of submodular subset selection */
16  for i = 1 to k do
17      S_d ← V\S;
18      α ← arg max_{α∈S_d} F(S ∪ {α}) ;                    /* Optimize the submodular value */
19      S ← S ∪ {α}
20  end
21  return S
```

---

Given a set $V = \left\{\mathbf{I}_1^M, \mathbf{I}_2^M, \cdots, \mathbf{I}_m^M\right\}$, we can follow Eq. 2 to search the interpretable region by selecting $k$ elements that maximize the value of the submodular function $\mathcal{F}(\cdot)$. The above problem can be effectively addressed by implementing a greedy search algorithm. Referring to related works (Mirzasoleiman et al., 2015; Wei et al., 2015), we use Algorithm 1 to optimize the value of the submodular function. Based on Lemma 1 and Lemma 2 we have proved that the function $\mathcal{F}(\cdot)$ is a submodular function. According to the theory of Nemhauser et al. (1978), we have:

**Theorem 1** ((Nemhauser et al., 1978)). *Let $S$ denotes the solution obtained by the greedy search approach, and $S^*$ denotes the optimal solution. If $\mathcal{F}(\cdot)$ is a submodular function, then the solution $S$ has the following approximation guarantee:*

$$\mathcal{F}(S) \geq \left(1 - \frac{1}{e}\right) \mathcal{F}(S^*),\tag{10}$$

*where $e$ is the base of natural logarithm.*

## 5 EXPERIMENTS

### 5.1 EXPERIMENTAL SETUP

**Datasets** We evaluate the proposed method on two face datasets Celeb-A (Liu et al., 2015) and VGG-Face2 (Cao et al., 2018), and a fine-grained dataset CUB-200-2011 (Welinder et al., 2010). Celeb-A dataset includes $10,177$ IDs, we randomly select $2,000$ identities from Celeb-A's validation set, and one test face image for each identity is used to evaluate our method; the VGG-Face2 dataset includes $8,631$ IDs, we randomly select $2,000$ identities from VGG-Face2's validation set, and one test face image for each identity is used to evaluate our method; CUB-200-2011 dataset includes 200 bird species, we select 3 samples for each class that is correctly predicted by the model from the CUB-200-2011 validation set for 200 classes, a total of 600 images are used to evaluate the image attribution effect. Additionally, 2 incorrectly predicted samples from each class are selected, totaling 400 images, to evaluate our method for identifying the cause of model prediction errors.

Table 1: Deletion and Insertion AUC scores on the Celeb-A, VGG-Face2, and CUB-200-2011 validation sets.

| Method | Celeb-A | | VGGFace2 | | CUB-200-2011 | |
|---|---|---|---|---|---|---|
| | Deletion (↓) | Insertion (↑) | Deletion (↓) | Insertion (↑) | Deletion (↓) | Insertion (↑) |
| Saliency (Simonyan et al., 2014) | 0.1453 | 0.4632 | 0.1907 | 0.5612 | 0.0682 | 0.6585 |
| Saliency (w/ ours) | **0.1254** | **0.5465** | **0.1589** | **0.6287** | **0.0675** | **0.6927** |
| Grad-CAM (Selvaraju et al., 2020) | 0.2865 | 0.3721 | 0.3103 | 0.4733 | 0.0810 | 0.7224 |
| Grad-CAM (w/ ours) | **0.1549** | **0.4927** | **0.1982** | **0.5867** | **0.0726** | **0.7231** |
| LIME (Ribeiro et al., 2016) | 0.1484 | 0.5246 | 0.2034 | 0.6185 | 0.1070 | 0.6812 |
| LIME (w/ ours) | **0.1366** | **0.5496** | **0.1653** | **0.6314** | **0.0941** | **0.6994** |
| Kernel Shap (Lundberg & Lee, 2017) | 0.1409 | 0.5246 | 0.2119 | 0.6132 | 0.1016 | 0.6763 |
| Kernel Shap (w/ ours) | **0.1352** | **0.5504** | **0.1669** | **0.6314** | **0.0951** | **0.6920** |
| RISE (Petsiuk et al., 2018) | 0.1444 | 0.5703 | 0.1375 | 0.6530 | 0.0665 | 0.7193 |
| RISE (w/ ours) | **0.1264** | **0.5719** | **0.1346** | **0.6548** | **0.0630** | **0.7245** |
| HSIC-Attribution (Novello et al., 2022) | 0.1151 | 0.5692 | 0.1317 | 0.6694 | 0.0647 | 0.6843 |
| HSIC-Attribution (w/ ours) | **0.1054** | **0.5752** | **0.1304** | **0.6705** | **0.0613** | **0.7262** |

**Evaluation Metric**   We use Deletion and Insertion AUC scores (Petsiuk et al., 2018) to evaluate the faithfulness of our method in explaining model predictions. To evaluate the ability of our method to search for causes of model prediction errors, we use the model's highest confidence in correct class predictions over different search ranges as the evaluation metric.

**Baselines**   We select the following popular attribution algorithm for calculating the prior saliency map in Sec. 4.1, and also for comparison methods, including the white-box-based methods: Saliency (Simonyan et al., 2014), Grad-CAM (Selvaraju et al., 2020), Grad-CAM++ (Chattopadhay et al., 2018), Score-CAM (Wang et al., 2020), and the black-box-based methods: LIME (Ribeiro et al., 2016), Kernel Shap (Lundberg & Lee, 2017), RISE (Petsiuk et al., 2018) and HSIC-Attribution (Novello et al., 2022).

**Implementation Details**   Please see Appendix C for details.

### 5.2   FAITHFULNESS ANALYSIS

To highlight the superiority of our method, we evaluate its faithfulness, which gauges the consistency of the generated explanations with the deep model's decision-making process (Li et al., 2021a). We use Deletion and Insertion AUC scores to form the evaluation metric, which measures the decrease (or increase) in class probability as important pixels (given by the saliency map) are gradually removed (or increased) from the image. Since our method is based on greedy search, we can determine the importance of the divided regions according to the order in which they are searched. We can evaluate the faithfulness of the method by comparing the Deletion or Insertion scores under different image perturbation scales (i.e. different $k$ sizes) with other attribution methods. We set the search subset size $k$ to be the same as the set size $m$, and adjust the size of the divided regions according to the order of the returned elements. The image is perturbed to calculate the value of Deletion or Insertion of our method.

Table 1 shows the results on the Celeb-A, VGG-Face2, and CUB-200-2011 validation sets. We find that no matter which attribution method's saliency map is used as the baseline, our method can improve its Deletion and Insertion scores. The performance of our approach is also directly influenced by the sophistication of the underlying attribution algorithm, with more advanced algorithms leading to superior results. The state-of-the-art method, HSIC-Attribution (Novello et al., 2022), shows improvements when combined with our method. On the Celeb-A dataset, our method reduces its Deletion score by 8.4% and improves its Insertion score by 1.1%. Similarly, on the CUB-200-2011 dataset, our method enhances the HSIC-Attribution method by reducing its Deletion score by 5.3% and improving its Insertion score by 6.1%. On the VGG-Face2 dataset, our method enhances the HSIC-Attribution method by reducing its Deletion score by 1.0% and improving its Insertion score by 0.2%, this slight improvement may be caused by the large number of noisy images in the VGG-Face2 dataset. It is precise because we reformulate the image attribution problem into a subset selection problem that we can alleviate the insufficient fine-grainiess of the attribution region, thus improving the faithfulness of existing attribution algorithms. Our method exhibits significant improvements across a spectrum of vision tasks and attribution algorithms, thereby highlighting its extensive generalizability. Please see Appendix H for visualizations of our method on these datasets.

Table 2: Evaluation of discovering the cause of incorrect predictions.

| Method | Average highest confidence (↑) | | | | Insertion (↑) |
|---|---|---|---|---|---|
| | (0-25%) | (0-50%) | (0-75%) | (0-100%) | |
| Grad-CAM++ (Chattopadhay et al., 2018) | 0.1988 | 0.2447 | 0.2544 | 0.2647 | 0.1094 |
| Grad-CAM++ (w/ ours) | **0.2424** | **0.3575** | **0.3934** | **0.4193** | **0.1672** |
| Score-CAM (Wang et al., 2020) | 0.1896 | 0.2323 | 0.2449 | 0.2510 | 0.1073 |
| Score-CAM (w/ ours) | **0.2491** | **0.3395** | **0.3796** | **0.4082** | **0.1622** |
| HSIC-Attribution (Novello et al., 2022) | 0.1709 | 0.2091 | 0.2250 | 0.2493 | 0.1446 |
| HSIC-Attribution (w/ ours) | **0.2430** | **0.3519** | **0.3984** | **0.4513** | **0.1772** |
| Patch 10×10 | **0.2020** | **0.4065** | **0.4908** | **0.5237** | 0.1519 |

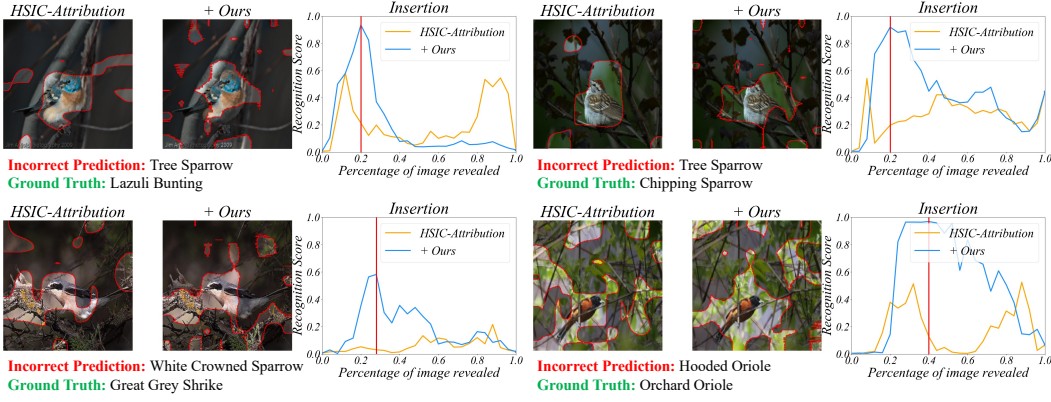

Figure 3: Visualization of the method for discovering what causes model prediction errors. The Insertion curve shows the correlation between the searched region and ground truth class prediction confidence. The highlighted region matches the searched region indicated by the red line in the curve, and the dark region is the error cause identified by the method.

## 5.3 DISCOVER THE CAUSES OF INCORRECT PREDICTIONS

In this section, we analyze the images that are misclassified in the CUB-200-2011 validation set and try to discover the causes for their misclassification. We use Grad-CAM++, Score-CAM, and HSIC-Attribution as baseline. We specify its ground truth class and use our attribution algorithm to discover regions where the model's response improves on that class as much as possible. We evaluate the performance of each method using the highest confidence score found within different search ranges. For instance, we determine the highest category prediction confidence that the algorithm can discover by searching up to 25% of the image region (i.e., set $k = 0.25m$). We also introduce the Insertion AUC score as an evaluation metric.

Table 2 shows the results for samples that were misclassified in the CUB-200-2011 validation set, we find that our method achieves significant improvements. The average highest confidence score searched by our method in any search interval is higher than the baseline. In the global search interval (0-100%), the average highest confidence level searched by our method is improved by 58.4% on Grad-CAM++, 62.6% on Score-CAM, and 81.0% on HSIC-Attribution. Its Insertion score has also been significantly improved, increasing by 52.8% on Grad-CAM++, 51.2% on Score-CAM, and 18.4% on HSIC-Attribution. In another scenario, our method does not rely on the saliency map of the attribution algorithm as a priori. Instead, it divides the image into $N \times N$ patches, where each element in the set contains only one patch. In this case, our method achieves a higher average highest confidence score, in the global search interval (0-100%), our method (divide the image into $10 \times 10$ patches) is 97.8% better than Grad-CAM++, 108.6% better than Score-CAM, and 110.1% better than HSIC-Attribution. Fig. 3 shows some visualization results, we compare our method with the SOTA method HSIC-Attribution. The Insertion curve represents the relationship between the region searched by the methods and the ground truth class prediction confidence. We find that our method can search for regions with higher confidence scores predicted by the model than the HSIC-Attribution method with a small percentage of the searched image region. The highlighted region shown in the figure can be considered as the cause of the correct prediction of the model, while the dark region is the cause of the incorrect prediction of the model. This also demonstrates that our method can achieve higher interpretability with fewer fine-grained regions. For further experiments and analysis of this method across various network backbones, please refer to Appendix D.

Table 3: Ablation study on components of different score functions of submodular function on the Celeb-A, and CUB-200-2011 validation sets.

| Submodular Function | | | | Celeb-A | | CUB-200-2011 | |
|---|---|---|---|---|---|---|---|
| Conf. Score (Equation 4) | Eff. Score (Equation 6) | Cons. Score (Equation 7) | Colla. Score (Equation 8) | Deletion (↓) | Insertion (↑) | Deletion (↓) | Insertion (↑) |
| ✓ | ✗ | ✗ | ✗ | 0.3161 | 0.1795 | 0.3850 | 0.3455 |
| ✗ | ✓ | ✗ | ✗ | 0.1211 | 0.5615 | 0.0835 | 0.6383 |
| ✗ | ✗ | ✓ | ✗ | 0.2849 | 0.2291 | 0.1019 | 0.6905 |
| ✗ | ✗ | ✗ | ✓ | 0.1591 | 0.3053 | 0.0771 | 0.5409 |
| ✓ | ✓ | ✗ | ✗ | 0.1075 | 0.5714 | 0.0865 | 0.6624 |
| ✗ | ✓ | ✓ | ✗ | 0.1082 | 0.5692 | 0.0750 | 0.7111 |
| ✗ | ✗ | ✓ | ✓ | 0.1558 | 0.3617 | 0.0641 | 0.7181 |
| ✗ | ✓ | ✓ | ✓ | 0.1074 | 0.5735 | 0.0632 | 0.7169 |
| ✓ | ✗ | ✓ | ✓ | 0.1993 | 0.2616 | 0.0623 | 0.7227 |
| ✓ | ✓ | ✗ | ✓ | 0.1067 | 0.5712 | 0.0651 | 0.6753 |
| ✓ | ✓ | ✓ | ✗ | 0.1088 | 0.5750 | 0.0811 | 0.7090 |
| ✓ | ✓ | ✓ | ✓ | **0.1054** | **0.5752** | **0.0613** | **0.7262** |

Table 4: Impact on whether to use a priori attribution map.

| Method | Divided set size $m$ | Celeb-A | | CUB-200-2011 | |
|---|---|---|---|---|---|
| | | Deletion (↓) | Insertion (↑) | Deletion (↓) | Insertion (↑) |
| Patch 7×7 | 49 | 0.1493 | 0.5642 | 0.1061 | 0.6903 |
| Patch 10×10 | 100 | 0.1365 | 0.5459 | 0.1024 | 0.6159 |
| Patch 14×14 | 196 | 0.1284 | 0.5562 | 0.0853 | 0.5805 |
| +HSIC-Attribution | 25 | **0.1054** | **0.5752** | **0.0613** | **0.7262** |

## 5.4 ABLATION STUDY

**Components of Submodular Function** We analyze the impact of various submodular function-based score functions on the results for both the Celeb-A and CUB-200-2011 datasets, the saliency map generated by the HSIC-Attribution method is used as a prior. As shown in Table 3, we observed that using a single score function within the submodular function limits attribution faithfulness. When these score functions are combined in pairs, the faithfulness of our method will be further improved. We demonstrate that removing any of the four score functions leads to deteriorated Deletion and Insertion scores, confirming the validity of these score functions. We find that the effectiveness score is key for the Celeb-A dataset, while the collaboration score is more important for the CUB-200-2011 dataset.

**Whether to Use a Priori Attribution Map** Our method uses existing attribution algorithms as priors when dividing the image into sub-regions. Table 4 shows the results, we directly divide the image into $N \times N$ patches without using a priori attribution map. Each element in the set only retains the pixel value of one patch area, and other areas are filled with values 0. We can observe that, whether on the Celeb-A dataset or the CUB-200-2011 dataset, the more patches are divided, the Deletion AUC score increases, and the fewer patches are divided, the Insertion AUC score increases. However, no matter how the number of patches is changed, we find that introducing a prior saliency map generated by the HSIC attribution method works best, in terms of Insertion or Deletion scores. Therefore, we believe that introducing a prior saliency map to assist in dividing image areas is effective and can improve the computational efficiency of the algorithm.

## 6 CONCLUSION

In this paper, we propose a new method that reformulates the attribution problem as a submodular subset selection problem. To address the lack of attention to local regions, we construct a novel submodular function to discover more accurate fine-grained interpretation regions. Specifically, four different constraints implemented on sub-regions are formulated together to evaluate the significance of various subsets, i.e., confidence, effectiveness, consistency, and collaboration scores. The proposed method has outperformed state-of-the-art methods on two face datasets (Celeb-A and VGG-Face2) and a fine-grained dataset (CUB-200-2011). Experimental results show that the proposed method can improve the Deletion and Insertion scores for the correctly predicted samples. While for incorrectly predicted samples, our method excels in identifying the reasons behind the model's decision-making errors.

ACKNOWLEDGMENTS

This work was supported in part by the National Key R&D Program of China (Grant No. 2022ZD0118102), in part by the National Natural Science Foundation of China (No. 62372448, 62025604, 62306308), and part by Beijing Natural Science Foundation (No. L212004).

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

# A  NOTATIONS

The notations used throughout this article are summarized in Table 5.

Table 5: Some important notations used in this paper.

| Notation | Description |
|---|---|
| $\mathbf{I}$ | an input image |
| $\mathbf{I}^M$ | a sub-region into which $\mathbf{I}$ is divided |
| $V$ | a finite set of divided sub-regions |
| $S$ | a subset of $V$ |
| $\alpha$ | an element from $V \setminus S$ |
| $k$ | the size of the $S$ |
| $\mathcal{F}(\cdot)$ | a function that maps a set to a value |
| $\mathcal{A}$ | saliency map calculated by attribution algorithms |
| $N \times N$ | the number of divided patches of $\mathbf{I}$ |
| $m$ | the number of sub-regions into which the image $\mathbf{I}$ is divided |
| $d$ | the number of patches in $\mathbf{I}^M$ |
| $\mathbf{x}$ | an input sample |
| $\mathbf{y}$ | the one-hot label |
| $u$ | the predictive uncertainty ranges from 0 to 1 |
| $F_u(\cdot)$ | a deep evidential network for calculating the $u$ of $\mathbf{x}$ |
| $F(\cdot)$ | a traditional network encoder |
| $K$ | the number of classes |
| $\text{dist}(\cdot, \cdot)$ | a function to calculate the distance between two feature vectors |
| $\mathbf{f}_s$ | the target semantic feature vector |

# B  THEORY PROOF

## B.1  PROOF OF LEMMA 1

*Proof.* Consider two sub-sets $S_a$ and $S_b$ in set $V$, where $S_a \subseteq S_b \subseteq V$. Given an element $\alpha$, where $\alpha = V \setminus S_b$. The necessary and sufficient conditions for the function $\mathcal{F}(\cdot)$ to satisfy the submodular property are:

$$\mathcal{F}\left(S_a \cup \{\alpha\}\right) - \mathcal{F}\left(S_a\right) \geq \mathcal{F}\left(S_b \cup \{\alpha\}\right) - \mathcal{F}\left(S_b\right). \tag{11}$$

For Eq. 4, assuming that the individual element $\alpha$ of the collection division is relatively small, according to the Taylor decomposition (Montavon et al., 2017), we can locally approximate $F_u(S_a + \alpha) = F_u(S_a) + \nabla F_u(S_a) \cdot \alpha$. Thus:

$$
\begin{aligned}
s_{\text{conf.}}\left(S_a + \alpha\right) - s_{\text{conf.}}\left(S_a\right) &= \frac{K}{\exp\left(F_u\left(S_a\right)\right) + K} - \frac{K}{\exp\left(F_u\left(S_a + \alpha\right)\right) + K} \\
&= \frac{K}{\exp\left(F_u\left(S_a\right)\right) + K} - \frac{K}{\exp\left(F_u\left(S_a\right)\right)\exp\left(\nabla F_u\left(S_a\right) \cdot \alpha\right) + K},
\end{aligned}
\tag{12}
$$

since $S_a \cap \alpha = \varnothing$, $S_a$ and $\alpha$ do not overlap in the image space, and $\alpha$ is small. Therefore, we can regard $\nabla F_u(S_a) \cdot \alpha \simeq 0$. Follow up:

$$
\begin{aligned}
s_{\text{conf.}}\left(S_a + \alpha\right) - s_{\text{conf.}}\left(S_a\right) &\simeq \frac{K}{\exp\left(F_u\left(S_a\right)\right) + K} - \frac{K}{\exp\left(F_u\left(S_a\right)\right)\exp\left(\mathbf{0}\right) + K} \\
&= 0_+,
\end{aligned}
\tag{13}
$$

and in the same way, $s_{\text{conf.}}\left(S_b + \alpha\right) - s_{\text{conf.}}\left(S_b\right) \simeq 0_+$. We have:

$$s_{\text{conf.}}\left(S_a + \alpha\right) - s_{\text{conf.}}\left(S_a\right) - \left(s_{\text{conf.}}\left(S_b + \alpha\right) - s_{\text{conf.}}\left(S_b\right)\right) \approx 0. \tag{14}$$

For Eq. 6, when a new element $\alpha$ is added to the set $S_a$, the minimum distance between elements in $S_a$ and other elements may be further reduced, i.e., for any element $s_i \in S_a$, we have $\min_{s_j \in S_a \cup \{\alpha\}, s_j \neq s_i} \text{dist}\left(F\left(s_i\right), F\left(s_j\right)\right) \leq \min_{s_j \in S_a, s_j \neq s_i} \text{dist}\left(F\left(s_i\right), F\left(s_j\right)\right)$. Thus:

$$
\begin{aligned}
s_{\text{eff.}}\left(S_a \cup \{\alpha\}\right) &= \min_{s_i \in S_a} \text{dist}\left(F\left(\alpha\right), F\left(s_i\right)\right) + \sum_{s_i \in S_a} \min_{s_j \in S_a \cup \{\alpha\}, s_j \neq s_i} \text{dist}\left(F\left(s_i\right), F\left(s_j\right)\right) \\
&= \min_{s_i \in S_a} \text{dist}\left(F\left(\alpha\right), F\left(s_i\right)\right) + \sum_{s_i \in S_a} \min_{s_j \in S_a, s_j \neq s_i} \text{dist}\left(F\left(s_i\right), F\left(s_j\right)\right) - \varepsilon_a,
\end{aligned}
\tag{15}
$$

where $\varepsilon_a$ is a constant, which is the sum of the minimum distance reductions of the elements in the original $S_a$ after $\alpha$ is added. Then, we have:

$$
s_{\text{eff.}}\left(S_a \cup \{\alpha\}\right) - s_{\text{eff.}}\left(S_a\right) = \min_{s_i \in S_a} \text{dist}\left(F\left(\alpha\right), F\left(s_i\right)\right) - \varepsilon_a,
\tag{16}
$$

and in the same way,

$$
s_{\text{eff.}}\left(S_b \cup \{\alpha\}\right) - s_{\text{eff.}}\left(S_b\right) = \min_{s_i \in S_b} \text{dist}\left(F\left(\alpha\right), F\left(s_i\right)\right) - \varepsilon_b,
\tag{17}
$$

since $S_a \subseteq S_b$, the minimum distance between alpha and elements in $S_b \setminus S_a$ may be smaller than the minimum distance between alpha and elements in $S_a$, thus,

$$
\min_{s_i \in S_a} \text{dist}\left(F\left(\alpha\right), F\left(s_i\right)\right) \geq \min_{s_i \in S_b} \text{dist}\left(F\left(\alpha\right), F\left(s_i\right)\right),
$$

since there are more elements in $S_b$ than in $S_a$, more elements in $S_b$ have the shortest distance from $\alpha$, that, $\varepsilon_b \geq \varepsilon_a$. Therefore, we have:

$$
s_{\text{eff.}}\left(S_a \cup \{\alpha\}\right) - s_{\text{eff.}}\left(S_a\right) \geq s_{\text{eff.}}\left(S_b \cup \{\alpha\}\right) - s_{\text{eff.}}\left(S_b\right).
\tag{18}
$$

For Eq. 7, let $G\left(S_a\right) = F\left(S_a\right) \cdot \boldsymbol{f}_s$. Assuming that the individual element $\alpha$ of the collection division is relatively small, according to the Taylor decomposition, we can locally approximate $G\left(S_a + \alpha\right) = G\left(S_a\right) + \nabla G\left(S_a\right) \cdot \alpha$. Assuming that the searched $\alpha$ is valid, i.e., $\nabla G\left(S_a\right) > 0$. Thus:

$$
\begin{aligned}
s_{\text{cons.}}\left(S_a + \alpha, \boldsymbol{f}_s\right) - s_{\text{cons.}}\left(S_a, \boldsymbol{f}_s\right) &= \frac{G(S_a) + \nabla G\left(S_a\right) \cdot \alpha}{\|F(S_a) + \nabla F\left(S_a\right) \cdot \alpha\| \|\boldsymbol{f}_s\|} - \frac{G(S_a)}{\|F(S_a)\| \|\boldsymbol{f}_s\|} \\
&\simeq \frac{\nabla G\left(S_a\right) \cdot \alpha}{\|F(S_a)\| \|\boldsymbol{f}_s\|},
\end{aligned}
\tag{19}
$$

since $S_a \cap \alpha = \varnothing$, $S_a$ and $\alpha$ do not overlap in the image space, and $\alpha$ is small, $\nabla G\left(S_a\right) \cdot \alpha$ is small. Then, we have:

$$
s_{\text{cons.}}\left(S_a + \alpha, \boldsymbol{f}_s\right) - s_{\text{cons.}}\left(S_a, \boldsymbol{f}_s\right) - \left(s_{\text{cons.}}\left(S_b + \alpha, \boldsymbol{f}_s\right) - s_{\text{cons.}}\left(S_b, \boldsymbol{f}_s\right)\right) \approx 0.
\tag{20}
$$

For Eq. 8, let $G\left(\mathbf{I} - S_a\right) = F\left(\mathbf{I} - S_a\right) \cdot \boldsymbol{f}_s$. Assuming that the individual element $\alpha$ of the collection division is relatively small, according to the Taylor decomposition, we can locally approximate $G\left(\mathbf{I} - S_a - \alpha\right) = G\left(\mathbf{I} - S_a\right) - \nabla G\left(\mathbf{I} - S_a\right) \cdot \alpha$. Assuming that the searched alpha is valid, i.e., $\nabla G\left(\mathbf{I} - S_a\right) > 0$. Thus:

$$
\begin{aligned}
s_{\text{colla.}}\left(S_a + \alpha, \mathbf{I}, \boldsymbol{f}_s\right) - s_{\text{colla.}}\left(S_a, \mathbf{I}, \boldsymbol{f}_s\right) &= \frac{G\left(\mathbf{I} - S_a\right)}{\|F\left(\mathbf{I} - S_a\right)\| \|\boldsymbol{f}_s\|} - \frac{G\left(\mathbf{I} - S_a\right) - \nabla G\left(\mathbf{I} - S_a\right) \cdot \alpha}{\|F\left(\mathbf{I} - S_a - \alpha\right)\| \|\boldsymbol{f}_s\|} \\
&\simeq \frac{\nabla G\left(\mathbf{I} - S_a\right) \cdot \alpha}{\|F\left(\mathbf{I} - S_a\right)\| \|\boldsymbol{f}_s\|},
\end{aligned}
\tag{21}
$$

since $\alpha$ is small, $\nabla G\left(\mathbf{I} - S_a\right) \cdot \alpha$ is small. Then, we have:

$$
s_{\text{colla.}}\left(S_a + \alpha, \mathbf{I}, \boldsymbol{f}_s\right) - s_{\text{colla.}}\left(S_a, \mathbf{I}, \boldsymbol{f}_s\right) - \left(s_{\text{colla.}}\left(S_b + \alpha, \mathbf{I}, \boldsymbol{f}_s\right) - s_{\text{colla.}}\left(S_b, \mathbf{I}, \boldsymbol{f}_s\right)\right) \approx 0.
\tag{22}
$$

Combining Eq. 14, 18, 20, and 22 we can get:

$$
\mathcal{F}\left(S_a \cup \{\alpha\}\right) - \mathcal{F}\left(S_a\right) \geq \mathcal{F}\left(S_b \cup \{\alpha\}\right) - \mathcal{F}\left(S_b\right),
\tag{23}
$$

hence, we can prove that Eq. 9 is a submodular function. $\qquad\square$

## B.2   PROOF OF LEMMA 2

*Proof.* Consider a subset $S$, given an element $\alpha$, assuming that $\alpha$ is contributing to interpretation. The necessary and sufficient conditions for the function $\mathcal{F}(\cdot)$ to satisfy the property of monotonically non-decreasing is:

$$\mathcal{F}(S \cup \{\alpha\}) - \mathcal{F}(S) > 0, \tag{24}$$

where, for Eq. 4:

$$s_{\text{conf.}}(S + \alpha) - s_{\text{conf.}}(S) = \frac{K}{\exp(F_u(S)) + K} - \frac{K}{\exp(F_u(S))\exp(\nabla F_u(S) \cdot \alpha) + K}, \tag{25}$$

since $\alpha$ is contributing to interpretation, $\nabla F_u(S) > 0$, and $\exp(\nabla F_u(S) \cdot \alpha) > 1$, thus:

$$s_{\text{conf.}}(S + \alpha) - s_{\text{conf.}}(S) > 0. \tag{26}$$

For Eq. 6,

$$s_{\text{eff.}}(S \cup \{\alpha\}) - s_{\text{eff.}}(S) = \min_{s_i \in S} \text{dist}(F(\alpha), F(s_i)) - \varepsilon,$$

since effective element $\alpha$ are selected as much as possible, the value $\varepsilon$ will be small,

$$s_{\text{eff.}}(S \cup \{\alpha\}) - s_{\text{eff.}}(S) \simeq \min_{s_i \in S} \text{dist}(F(\alpha), F(s_i)) > 0. \tag{27}$$

For Eq. 7, assuming that the searched $\alpha$ is valid,

$$s_{\text{cons.}}(S + \alpha, \boldsymbol{f}_s) - s_{\text{cons.}}(S, \boldsymbol{f}_s) \simeq \frac{\nabla G(S) \cdot \alpha}{\|F(S)\|\|\boldsymbol{f}_s\|} > 0, \tag{28}$$

likewise, for Eq. 8,

$$s_{\text{colla.}}(S + \alpha, \mathbf{I}, \boldsymbol{f}_s) - s_{\text{colla.}}(S, \mathbf{I}, \boldsymbol{f}_s) \simeq \frac{\nabla G(\mathbf{I} - S) \cdot \alpha}{\|F(\mathbf{I} - S)\|\|\boldsymbol{f}_s\|} > 0. \tag{29}$$

Combining Eq. 26, 27, 28, and 29 we can get:

$$\mathcal{F}(S \cup \{\alpha\}) - \mathcal{F}(S) > 0, \tag{30}$$

hence, we can prove that Eq. 9 is monotonically non-decreasing.

$\square$

## C   IMPLEMENTATION DETAILS

We primarily employed CNN-based architectures to explain our models. For the face datasets, we evaluated recognition models that were trained using the ResNet-101 (He et al., 2016) architecture and the ArcFace (Deng et al., 2019) loss function, with an input size of $112 \times 112$ pixels. For the CUB-200-2011 dataset, we evaluated a recognition model trained on the ResNet-101 architecture with a cross-entropy loss function and an input size of $224 \times 224$ pixels. It's worth noting that all the uncertainty models denoted as $F_u(\cdot)$ in Sec. 4.2 were trained using the ResNet101 architecture. Given that the face recognition task primarily involves face verification, we use the target semantic feature $\boldsymbol{f}_s$ in functions $s_{\text{cons.}}(S, \boldsymbol{f}_s)$ and $s_{\text{colla.}}(S, \mathbf{I}, \boldsymbol{f}_s)$ to adopt features computed from the original image using a pre-trained feature extractor. For the CUB-200-2011 dataset, we directly implement the fully connected layer of the classifier for a specified class. For the two face datasets, we set $N = 28$ and $m = 98$. For the CUB-200-2011 dataset, we set $N = 10$ and $m = 25$. We conduct our experiments using the Xplique [1] repository, which offers baseline methods and evaluation tools. To evaluate the Deletion and Insertion AUC scores, we set $k$ to be the same as the set size $m$ to evaluate the importance ranking of different sub-regions. These experiments were performed on an NVIDIA 3090 GPU.

---

[1]Xplique: https://github.com/deel-ai/xplique

Table 6: Evaluation on discovering the cause of incorrect predictions for different network backbones.

| Backbone | Method | Average highest confidence (↑) | | | | Insertion (↑) |
|---|---|---|---|---|---|---|
| | | (0-25%) | (0-50%) | (0-75%) | (0-100%) | |
| VGGNet-19 (Simonyan & Zisserman, 2015) | Grad-CAM++ (Chattopadhay et al., 2018) | 0.1323 | 0.2130 | 0.2427 | 0.2925 | 0.1211 |
| | Grad-CAM++ (w/ ours) | **0.1595** | **0.2615** | **0.3521** | **0.4263** | **0.1304** |
| | Score-CAM (Wang et al., 2020) | 0.1349 | 0.2125 | 0.2583 | 0.3058 | 0.1057 |
| | Score-CAM (w/ ours) | **0.1649** | **0.2624** | **0.3452** | **0.4224** | **0.1186** |
| | HSIC-Attribution (Novello et al., 2022) | 0.1456 | 0.1743 | 0.1906 | 0.2483 | 0.1297 |
| | HSIC-Attribution (w/ ours) | **0.1745** | **0.2716** | **0.3477** | **0.4226** | **0.1365** |
| ResNet-101 (He et al., 2016) | Grad-CAM++ (Chattopadhay et al., 2018) | 0.1988 | 0.2447 | 0.2544 | 0.2647 | 0.1094 |
| | Grad-CAM++ (w/ ours) | **0.2424** | **0.3575** | **0.3934** | **0.4193** | **0.1672** |
| | Score-CAM (Wang et al., 2020) | 0.1896 | 0.2323 | 0.2449 | 0.2510 | 0.1073 |
| | Score-CAM (w/ ours) | **0.2491** | **0.3395** | **0.3796** | **0.4082** | **0.1622** |
| | HSIC-Attribution (Novello et al., 2022) | 0.1709 | 0.2091 | 0.2250 | 0.2493 | 0.1446 |
| | HSIC-Attribution (w/ ours) | **0.2430** | **0.3519** | **0.3984** | **0.4513** | **0.1772** |
| MobileNetV2 (Sandler et al., 2018) | Grad-CAM++ (Chattopadhay et al., 2018) | 0.1584 | 0.2820 | 0.3223 | 0.3462 | 0.1284 |
| | Grad-CAM++ (w/ ours) | **0.1680** | **0.3565** | **0.4615** | **0.5076** | **0.1759** |
| | Score-CAM (Wang et al., 2020) | 0.1574 | 0.2456 | 0.2948 | 0.3141 | 0.1195 |
| | Score-CAM (w/ ours) | **0.1631** | **0.3403** | **0.4283** | **0.4893** | **0.1667** |
| | HSIC-Attribution (Novello et al., 2022) | 0.1648 | 0.2190 | 0.2415 | 0.2914 | 0.1635 |
| | HSIC-Attribution (w/ ours) | **0.2460** | **0.4142** | **0.4913** | **0.5367** | **0.1922** |
| EfficientNetV2-M (Tan & Le, 2021) | Grad-CAM++ (Chattopadhay et al., 2018) | 0.2338 | 0.2549 | 0.2598 | 0.2659 | 0.1605 |
| | Grad-CAM++ (w/ ours) | **0.2502** | **0.3038** | **0.3146** | **0.3214** | **0.1795** |
| | Score-CAM (Wang et al., 2020) | 0.2126 | 0.2327 | 0.2375 | 0.2403 | 0.1572 |
| | Score-CAM (w/ ours) | **0.2442** | **0.2900** | **0.3029** | **0.3115** | **0.1745** |
| | HSIC-Attribution (Novello et al., 2022) | 0.2418 | 0.2561 | 0.2615 | 0.2679 | 0.1611 |
| | HSIC-Attribution (w/ ours) | **0.2616** | **0.3117** | **0.3235** | **0.3306** | **0.1748** |

## D  DIFFERENT NETWORK BACKBONE

We verified the effect of our method on more CNN network architectures, including VGGNet-19 (Simonyan & Zisserman, 2015), MobileNetV2 (Sandler et al., 2018), and EfficientNetV2-M (Tan & Le, 2021). We adopted the same settings as the original network, with an input size of $224 \times 224$ for VGGNet-19, $224 \times 224$ for MobileNetV2, and $384 \times 384$ for EfficientNetV2-M. We conduct the experiment under the same setting as those employed for ResNet-101. When employing VGGNet-19, our method outperforms the SOTA method HSIC-Attribution, achieving a 70.2% increase in the average highest confidence and a 5.2% improvement in the Insertion AUC score. While utilizing MobileNetV2, our method outperforms the SOTA method HSIC-Attribution, achieving an 84.2% increase in the average highest confidence and a 17.6% improvement in the Insertion AUC score. Similarly, with EfficientNetV2-M, our method surpasses HSIC-Attribution, achieving a 23.4% increase in the average highest confidence and an 8.5% improvement in the Insertion AUC score. This indicates the versatility and effectiveness of our method across various backbones. Note that the 400 incorrectly predicted samples used by these networks in the experiment are not exactly the same, given the differences across the networks.

## E  HYPERPARAMETER SENSITIVITY ANALYSIS

We explored the effects of different score function weighting on the CUB-200-2011 dataset. As shown in Table 7, we find that only when increasing the weight of Eq. 7, the Insertion AUC score will be slightly improved, and other indicators have no obvious effect. If the weight is set too high, it may cause other score functions to fail, resulting in a significant decline in indicator performance. Therefore, we do not need to pay too much attention to how to weigh these score functions. The default weight is set to 1.

## F  ABLATION ON MORE DETAILS

**Ablation on division sub-region number $N \times N$ and set size $m$**    In this section, we explore the impact of the size of the divided sub-region number $N \times N$ and the size of the set $m$ on attribution performance. The experimental settings are set by default unless otherwise specified. We conducted experiments on the Celeb-A dataset and the CUB-200-2011 dataset. For the Celeb-A dataset, as shown in Table 8, we found that the more granular the division on the face dataset, the better the attribution results can be achieved, however, the attribution effect at the pixel level will be relatively poor. For the face dataset, a better choice is set $N = 28$ and $m = 98$. On the CUB-200-2011

Table 7: Ablation study on the effects of different score functions on the CUB-200-2011 validation set.

| Conf. Score (Equation 4) | Eff. Score (Equation 6) | Cons. Score (Equation 7) | Colla. Score (Equation 8) | Deletion (↓) | Insertion (↑) |
|---|---|---|---|---|---|
| 1 | 1 | 1 | 1 | **0.0613** | 0.7262 |
| 20 | 1 | 1 | 1 | 0.1333 | 0.5979 |
| 50 | 1 | 1 | 1 | 0.2529 | 0.4681 |
| 100 | 1 | 1 | 1 | 0.3198 | 0.4059 |
| 1 | 20 | 1 | 1 | 0.0753 | 0.6725 |
| 1 | 50 | 1 | 1 | 0.0781 | 0.6614 |
| 1 | 100 | 1 | 1 | 0.0793 | 0.6537 |
| 1 | 1 | 20 | 1 | 0.0645 | **0.7276** |
| 1 | 1 | 50 | 1 | 0.0664 | **0.7276** |
| 1 | 1 | 100 | 1 | 0.0689 | 0.7221 |
| 1 | 1 | 1 | 20 | 0.0597 | 0.7174 |
| 1 | 1 | 1 | 50 | 0.0615 | 0.7001 |
| 1 | 1 | 1 | 100 | 0.0657 | 0.7010 |

Table 8: Ablation study on division sub-region number $N \times N$ and set size $m$ on the Celeb-A validation set. The input image size is $112 \times 112$, the base attribution method is HSIC-Attribution (Novello et al., 2022).

| Division method | Number of divided areas for a element | Collection size | Deletion (↓) | Insertion (↑) |
|---|---|---|---|---|
| Patch 28×28 | 8 | 98 | **0.1054** | 0.5752 |
| Patch 14×14 | 2 | 98 | 0.1158 | 0.5703 |
| Patch 14×14 | 4 | 49 | 0.1133 | **0.5757** |
| Patch 10×10 | 2 | 50 | 0.1288 | 0.5621 |
| Patch 10×10 | 4 | 25 | 0.1235 | 0.5645 |
| Patch 8×8 | 2 | 32 | 0.1276 | 0.5542 |
| Pixel 112×112 | 128 | 98 | 0.1258 | 0.5404 |
| Pixel 112×112 | 256 | 49 | 0.1148 | 0.5615 |
| Pixel 112×112 | 392 | 32 | 0.1136 | 0.5664 |

dataset, as shown in Table 9, the attribution effect at the pixel level is still relatively poor. Since the positions of birds in the CUB-200-2011 dataset vary greatly and the background is complex, the images are not suitable for too fine-grained division. For the CUB-200-2011 dataset, a better choice is to set $N = 10$ and $m = 4$.

**Ablation of score function on the attribution of incorrectly predicted samples**  Since the attribution of mispredicted samples requires specifying the correct category, we mainly discuss the impact of Eq. 7 and Eq. 8 on the attribution of mispredicted samples because these equations in-

Table 9: Ablation study on division sub-region number $N \times N$ and set size $m$ on the CUB-200-2011 validation set. The input image size is $224 \times 224$, and the base attribution method is HSIC-Attribution (Novello et al., 2022).

| Division method | Number of divided areas for a element | Collection size | Deletion (↓) | Insertion (↑) |
|---|---|---|---|---|
| Patch 14×14 | 2 | 98 | 0.0769 | 0.6343 |
| Patch 14×14 | 4 | 49 | 0.0652 | 0.6661 |
| Patch 10×10 | 2 | 50 | 0.0689 | 0.6917 |
| Patch 10×10 | 4 | 25 | **0.0613** | **0.7262** |
| Patch 8×8 | 2 | 32 | 0.0686 | 0.7079 |
| Pixel 224×224 | 256 | 49 | 0.0859 | 0.5818 |
| Pixel 224×224 | 392 | 32 | 0.0768 | 0.6398 |

Table 10: Ablation study on submodular function score components for incorrectly predicted samples in the CUB-200-2011 dataset.

| Method | Cons. Score (Equation 7) | Colla. Score (Equation 8) | Average highest confidence (↑) | | | | Insertion (↑) |
|---|---|---|---|---|---|---|---|
| | | | (0-25%) | (0-50%) | (0-75%) | (0-100%) | |
| Grad-CAM++ (w/ ours) | ✗ | ✓ | 0.0821 | 0.1547 | 0.1923 | 0.2303 | 0.1122 |
| Grad-CAM++ (w/ ours) | ✓ | ✗ | 0.1654 | 0.2888 | 0.3338 | 0.3611 | 0.1452 |
| Grad-CAM++ (w/ ours) | ✓ | ✓ | **0.2424** | **0.3575** | **0.3934** | **0.4193** | **0.1672** |
| Score-CAM (w/ ours) | ✗ | ✓ | 0.0742 | 0.1348 | 0.1835 | 0.2237 | 0.1072 |
| Score-CAM (w/ ours) | ✓ | ✗ | 0.1383 | 0.2547 | 0.3131 | 0.3402 | 0.1306 |
| Score-CAM (w/ ours) | ✓ | ✓ | **0.2491** | **0.3395** | **0.3796** | **0.4082** | **0.1622** |
| HSIC-Attribution (w/ ours) | ✗ | ✓ | 0.1054 | 0.1803 | 0.2177 | 0.2600 | 0.1288 |
| HSIC-Attribution (w/ ours) | ✓ | ✗ | 0.2394 | 0.3479 | 0.3940 | 0.4220 | 0.1645 |
| HSIC-Attribution (w/ ours) | ✓ | ✓ | **0.2430** | **0.3519** | **0.3984** | **0.4513** | **0.1772** |

Table 11: Ablation study on the effect of sub-region division size $N \times N$ in incorrect sample attribution.

| Method | Average highest confidence (↑) | | | | Insertion (↑) |
|---|---|---|---|---|---|
| | (0-25%) | (0-50%) | (0-75%) | (0-100%) | |
| HSIC-Attribution (Novello et al., 2022) | 0.1709 | 0.2091 | 0.2250 | 0.2493 | 0.1446 |
| Patch 5×5 | **0.2597** | 0.3933 | 0.4389 | 0.4514 | **0.1708** |
| Patch 6×6 | 0.2372 | 0.4025 | 0.4555 | 0.4720 | 0.1538 |
| Patch 7×7 | 0.2430 | **0.4289** | 0.4819 | 0.4985 | 0.1621 |
| Patch 8×8 | 0.1903 | 0.4005 | 0.4740 | 0.5043 | 0.1584 |
| Patch 10×10 | 0.2020 | 0.4065 | 0.4908 | 0.5237 | 0.1519 |
| Patch 12×12 | 0.1667 | 0.3816 | **0.4987** | **0.5468** | 0.1247 |

volve specifying categories. The network backbone we studied is ResNet-101, and 400 samples in the CUB-200-2011 test set that were misclassified by this network were used as research objects. As shown in Table 10, when any score function is removed, regardless of the imputation algorithm it is based on, the average highest confidence and Insertion AUC score will decrease, with Eq. 7 having the greatest impact.

**Ablation of division sub-region number $N \times N$ of incorrect sample attribution**    Without using a priori saliency map, we study the impact of different patch division numbers on the error sample attribution effect, where the number of patches $m = N \times N$. We explore the impact of six different division sizes on the results. As depicted in Table 11, our results indicate that dividing the image into more patches yields higher average highest confidence scores (0-100%). However, excessive division can introduce instability in the early search area (0-25%). In summary, without incorporating a priori saliency maps for division, opting for a 10x10 patch division followed by subset selection appears to be the optimal choice.

## G    EFFECTIVENESS

In this section, we discuss the processing time effectiveness of the proposed algorithm. The size $m$ of the divided set we are testing is 98, we sequentially test the processing time required by our algorithm under different settings of the number of search elements $k$.

As shown in Fig. 4, the more elements are searched, the longer times it takes, and the relationship is almost linear. To use the proposed method efficiently, it is better to control the divided set size $m$, the size of the search set can be controlled by controlling the number of divided patches $N$, or by controlling the parameter $d$ that controls the number of patches assigned to each element.

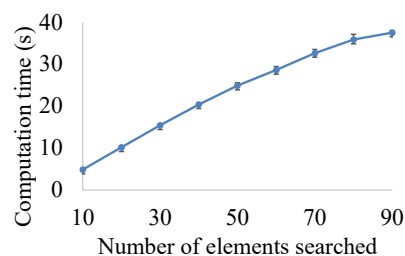

Figure 4: Processing time of our method.

# H VISUALIZING ATTRIBUTION OF CORRECTLY PREDICTED CLASSES

## H.1 CELEB-A

The visualization results compared with the HSIC-Attribution method are shown in Fig. 5. Our method has a higher Insertion AUC curve area and searches for the highest confidence with higher attribution results than the HSIC-Attribution method.

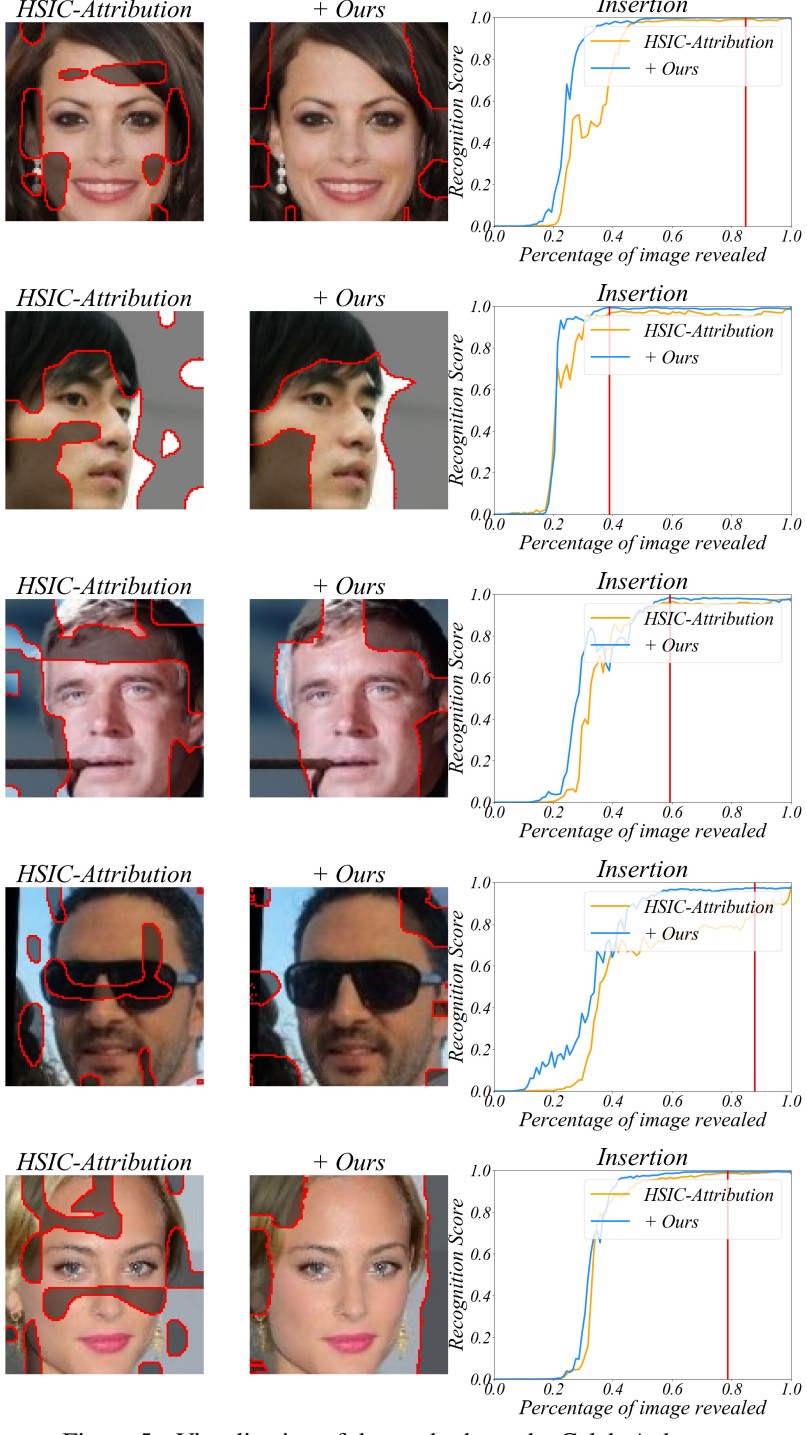

Figure 5: Visualization of the methods on the Celeb-A dataset.

## H.2 CUB-200-2011

The visualization results compared with the HSIC-Attribution method are shown in Fig. 6. Our method has a higher Insertion AUC curve area and searches for the highest confidence with higher attribution results than the HSIC-Attribution method.

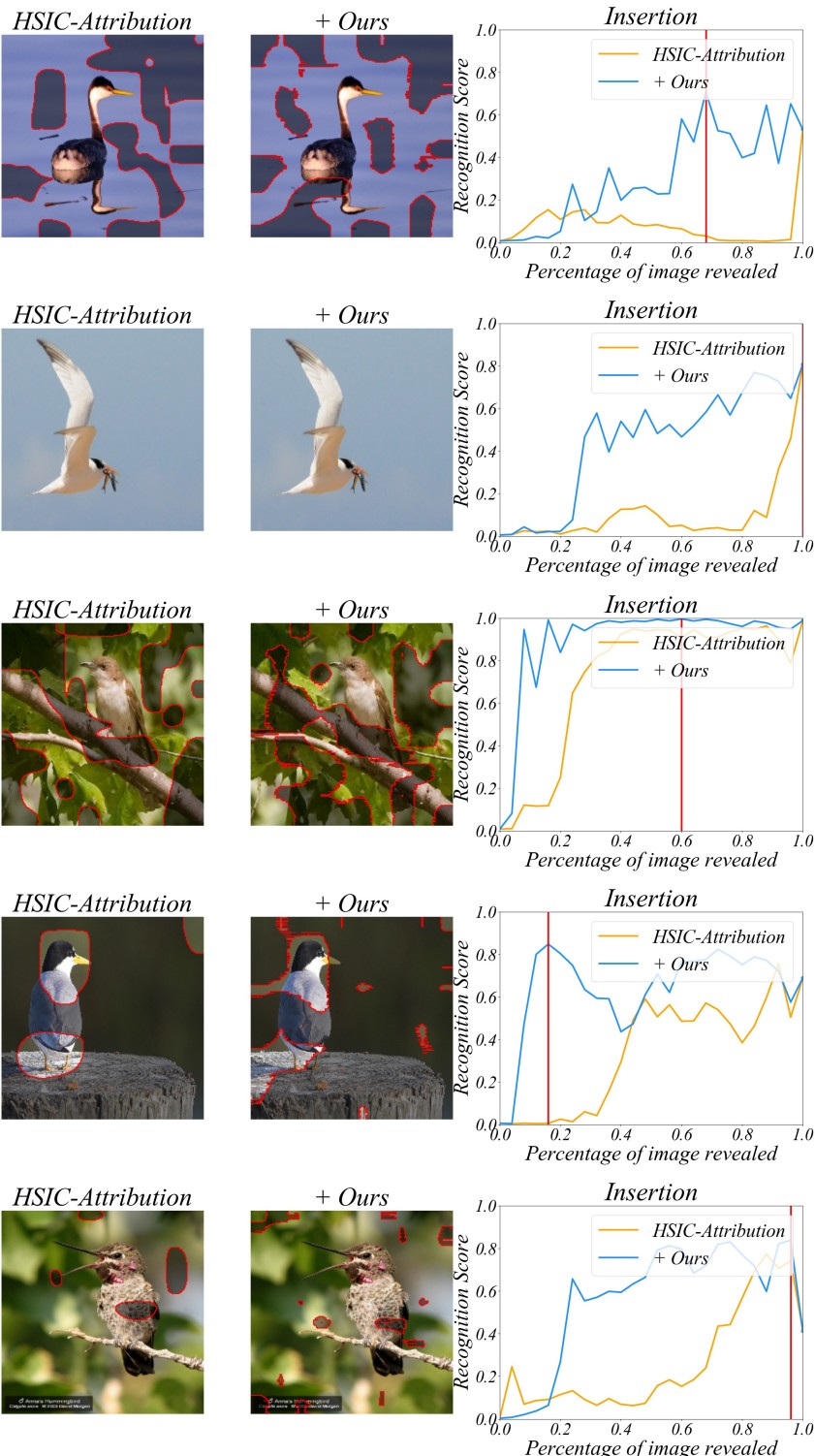

Figure 6: Visualization of the methods on the CUB-200-2011 dataset.

# I VISUALIZING ATTRIBUTION OF MISCLASSIFIED PREDICTION

Visualization of our method searches for causes of model prediction errors compared with the HSIC-Attribution method, as shown in Fig. 7. Our method has a higher Insertion AUC curve area and searches for the highest confidence with higher attribution results than the HSIC-Attribution method.

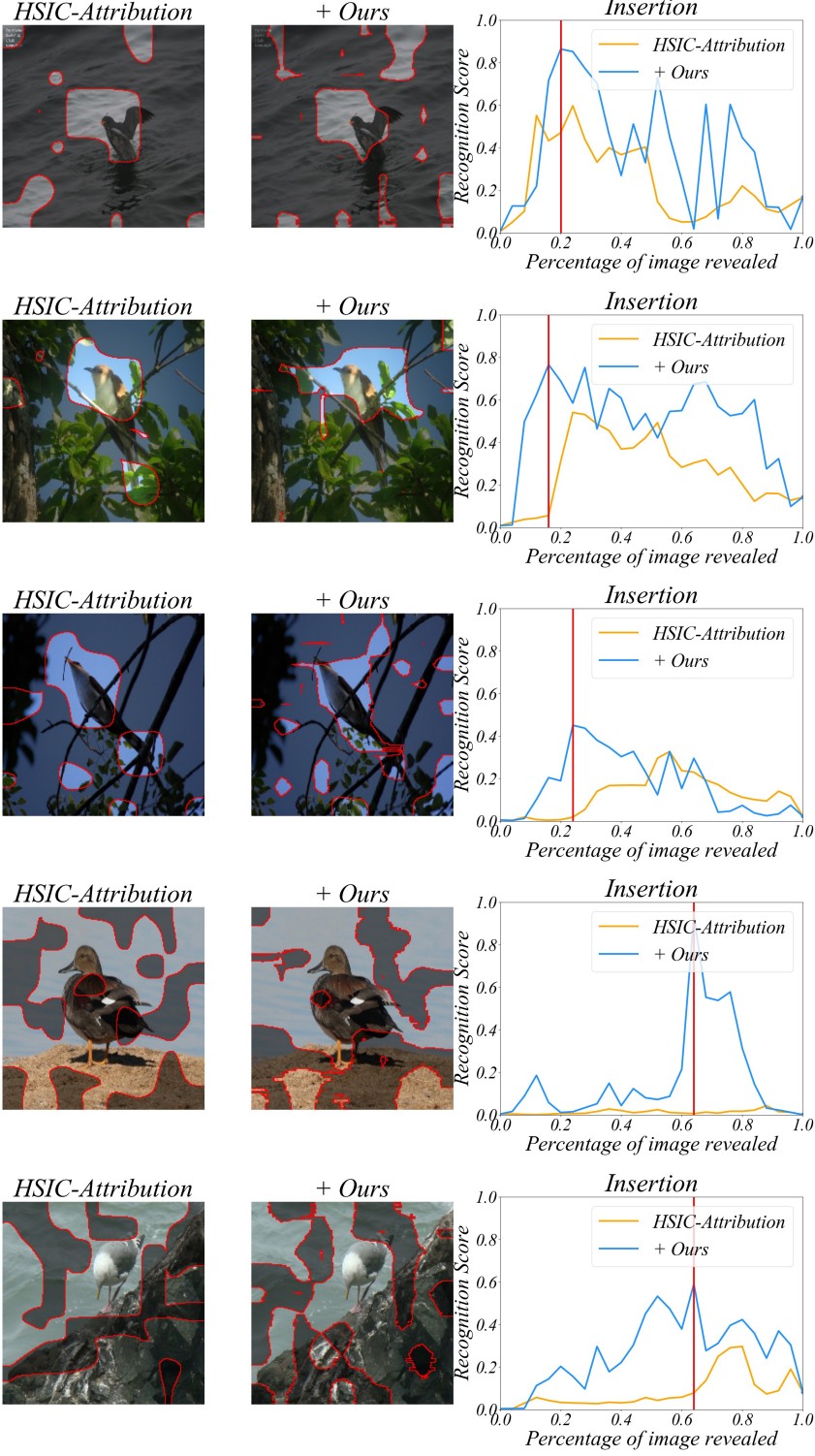

Figure 7: Visualization of the methods on CUB-200-2011 misclassified dataset.

## J    LIMITATION

The main limitation of our method is the computation time depending on the sub-region division approach. Specifically, we first divide the image into different subregions, and then we use the greed algorithm to search for the important subset regions. To accelerate the search process, we introduce the prior maps, which are computed based on the existing attribution methods. However, we observe that the performance of our attribution method is based on the scale of subregions, where the smaller regions would achieve much better accuracy as shown in the experiments. There exists a trade-off between the accuracy and the computation time. In the future, we will explore a better optimization strategy to solve this limitation.

