# OpenReview forum: "Less is More: Fewer Interpretable Region via Submodular Subset Selection"
_ICLR.cc/2024/Conference — ICLR 2024 oral_

### Official Review · Reviewer_2VHY · 2023-10-30

**Soundness:** 4 excellent
**Presentation:** 3 good
**Contribution:** 4 excellent
**Rating:** 8
**Confidence:** 4

**Summary:**

This paper applies the submodular subset selection theory to the image attribution method, and can effectively improve the attribution ability of the baseline saliency map. The authors verified the effectiveness of this method on the Celeb-A, VGGFace2 and CUB datasets. Experiments show that the method proposed in this article can obtain more effective explanations with fewer regions than the baseline method. In addition, this method has great advantages in searching for regions that lead to model prediction errors.

**Strengths:**

- It is very interesting and practical to use interpretability to find the reasons for model prediction errors, which is helpful for humans to discover model defects and assist in improvement.

- Different from perturbation-based and gradient-based methods, the author adopts a method based on searching image areas for image attribution and achieves better performance.

- The author conducted a detailed analysis and included more experimental results and visualization results in the appendix.

**Weaknesses:**

- In Section 5.3, for the discover the causes of incorrect predictions, the author only verified it on ResNet and achieved good quantitative results. It would be more convincing if the author could try to add some backbone, such as VGGNet.

- Why are the results of LIME and Kernel Shap under CUB data not reported in Table 1?

- In Table 2, the saliency map without a priori seems to be better than the method of adding a priori saliency map in terms of average highest confidence evaluation metric. Can the author add an ablation experiment to observe the impact of different partition sizes on the results without adding a priori saliency map, such as 8x8, 12x12, etc.

- In the introduction, “and a fine-grained dataset CUB-200-2011 (Welinder et al., 2010) datasets” -> “and a fine-grained dataset CUB-200-2011 (Welinder et al., 2010)”

- In Algorithm 1, the input k is not used, please check carefully.

**Questions:**

See weakness

---

> ### Author Response · Authors · 2023-11-20
> **Official Response to Reviewer 2VHY (1/3)**
>
> Thanks for your encouraging evaluations and constructive comments. We provide a detailed response, supplemented with additional experiments, to address your questions and concerns.
>
> ----
>
> **W1:** In Section 5.3, for the discover the causes of incorrect predictions, the author only verified it on ResNet and achieved good quantitative results. It would be more convincing if the author could try to add some backbone, such as VGGNet.
>
> **AW1:** Thanks for your helpful suggestion. We have added three different backbone architectures in this experiment to validate the generalization of our method. Besides the VGGNet, we also use two alternative and more advanced backbones: MobileNetV2 and EfficientNetV2-M.
>
> We conducted the experiment under the same setting as those employed for ResNet-101. When employing VGGNet-19, our method outperforms the SOTA method HSIC-Attribution, achieving a 70.2% increase in the average highest confidence and a 5.2% improvement in the Insertion AUC score. While utilizing MobileNetV2, our method outperforms the SOTA method HSIC-Attribution, achieving an 84.2% increase in the average highest confidence and a 17.6% improvement in the Insertion AUC score. Similarly, with EfficientNetV2-M,  our method surpasses HSIC-Attribution, achieving a 23.4% increase in the average highest confidence and an 8.5% improvement in the Insertion AUC score. This indicates the versatility and effectiveness of our method across various backbones. Note that the 400 incorrectly predicted samples used by VGGNet-19, MobileNetV2, and EfficientNetV2-M in the experiment are not exactly the same, given the differences across the networks.
>
> Due to the length of the paper, we have added these quantitative experiments in the appendix. Please see Appendix Section D and Table 6 of the revised version.
>
> Table 6-1. Evaluation of discovering the cause of incorrect predictions for VGGNet-19.
>
> | Method                     | Avg. highest conf. (0-25%) (↑) | Avg. highest conf. (0-50%) (↑) | Avg. highest conf. (0-75%) (↑) | Avg. highest conf. (0-100%) (↑) | Insertion ($\uparrow$) |
> | -------------------------- | :----------------------------: | :----------------------------: | :----------------------------: | :-----------------------------: | :--------------------: |
> | Grad-CAM++                 |             0.1323             |             0.2130             |             0.2427             |             0.2925              |         0.1211         |
> | Grad-CAM++ (w/ ours)       |           **0.1595**           |           **0.2615**           |           **0.3521**           |           **0.4263**            |       **0.1304**       |
> | Score-CAM                  |             0.1349             |             0.2125             |             0.2583             |             0.3058              |         0.1057         |
> | Score-CAM (w/ ours)        |           **0.1649**           |           **0.2624**           |           **0.3452**           |           **0.4224**            |       **0.1186**       |
> | HSIC-Attribution           |             0.1456             |             0.1743             |             0.1906             |             0.2483              |         0.1297         |
> | HSIC-Attribution (w/ ours) |           **0.1745**           |           **0.2716**           |           **0.3477**           |           **0.4226**            |       **0.1365**       |
>
> continue down

---

> ### Author Response · Authors · 2023-11-20
> **Official Response to Reviewer 2VHY (2/3)**
>
> Continue From Above
>
> Table 6-2. Evaluation of discovering the cause of incorrect predictions for MobileNetV2.
>
> | Method                     | Avg. highest conf. (0-25%) (↑) | Avg. highest conf. (0-50%) (↑) | Avg. highest conf. (0-75%) (↑) | Avg. highest conf. (0-100%) (↑) | Insertion ($\uparrow$) |
> | -------------------------- | :----------------------------: | :----------------------------: | :----------------------------: | :-----------------------------: | :--------------------: |
> | Grad-CAM++                 |             0.1584             |             0.2820             |             0.3223             |             0.3462              |         0.1284         |
> | Grad-CAM++ (w/ ours)       |           **0.1680**           |           **0.3565**           |           **0.4615**           |           **0.5076**            |       **0.1759**       |
> | Score-CAM                  |             0.1574             |             0.2456             |             0.2948             |             0.3141              |         0.1195         |
> | Score-CAM (w/ ours)        |           **0.1631**           |           **0.3403**           |           **0.4283**           |           **0.4893**            |       **0.1667**       |
> | HSIC-Attribution           |             0.1648             |             0.2190             |             0.2415             |             0.2914              |         0.1635         |
> | HSIC-Attribution (w/ ours) |           **0.2460**           |           **0.4142**           |           **0.4913**           |           **0.5367**            |       **0.1922**       |
>
> Table 6-3. Evaluation of discovering the cause of incorrect predictions for EfficientNetV2-M.
>
> | Method                     | Avg. highest conf. (0-25%) (↑) | Avg. highest conf. (0-50%) (↑) | Avg. highest conf. (0-75%) (↑) | Avg. highest conf. (0-100%) (↑) | Insertion ($\uparrow$) |
> | -------------------------- | :----------------------------: | :----------------------------: | :----------------------------: | :-----------------------------: | :--------------------: |
> | Grad-CAM++                 |             0.2338             |             0.2549             |             0.2598             |             0.2659              |         0.1605         |
> | Grad-CAM++ (w/ ours)       |           **0.2502**           |           **0.3038**           |           **0.3146**           |           **0.3214**            |       **0.1795**       |
> | Score-CAM                  |             0.2126             |             0.2327             |             0.2375             |             0.2403              |         0.1572         |
> | Score-CAM (w/ ours)        |           **0.2442**           |           **0.2900**           |           **0.3029**           |           **0.3115**            |       **0.1745**       |
> | HSIC-Attribution           |             0.2418             |             0.2561             |             0.2615             |             0.2679              |         0.1611         |
> | HSIC-Attribution (w/ ours) |           **0.2616**           |           **0.3117**           |           **0.3235**           |           **0.3306**            |       **0.1748**       |

---

> ### Author Response · Authors · 2023-11-20
> **Official Response to Reviewer 2VHY (3/3)**
>
> **W2:** Why are the results of LIME and Kernel Shap under CUB data not reported in Table 1?
>
> **AW2:** Thanks for your kind reminder. This limitation arises when using the Xplique library as both LIME and Kernel Shap methods are unable to calculate the saliency map for the ResNet-101 network trained on the CUB-200-2011 data set. This issue may be inherent to the Xplique library, leading us not to report relevant experiments.
>
> We have repeated this experiment based on other libraries, using the same network. The table below demonstrates the effectiveness of our method when applied to LIME and Kernel Shap on the CUB-200-2011 dataset. We have also included these experimental results in the revised manuscript.
>
> | Method                | Deletion ($\downarrow$) | Insertion ($\uparrow$) |
> | --------------------- | :---------------------: | :--------------------: |
> | LIME                  |         0.1070          |         0.6812         |
> | LIME (w/ ours)        |       **0.0941**        |       **0.6994**       |
> | Kernel Shap           |         0.1016          |         0.6763         |
> | Kernel Shap (w/ ours) |       **0.0951**        |       **0.6920**       |
>
> ----
>
> **W3:** In Table 2, the saliency map without a priori seems to be better than the method of adding a priori saliency map in terms of average highest confidence evaluation metric. Can the author add an ablation experiment to observe the impact of different partition sizes on the results without adding a priori saliency map, such as 8x8, 12x12, etc.
>
> **AW3:** Thanks for your helpful suggestion. We have added relevant experiments, specifically exploring the impact of six different division sizes on the results. As depicted in the table below, our results indicate that dividing the image into more patches yields higher average highest confidence scores (0-100%). However, excessive division can introduce instability in the early search area (0-25%). In summary, without incorporating a priori saliency maps for division, opting for a 10x10 patch division followed by subset selection appears to be the optimal choice.
>
> Due to the length of the paper, we have added these quantitative experiments in the appendix. Please see Appendix Section F and Table 11 of the revised version.
>
> Table 11. Ablation study on the effect of sub-region division size $N \times N$ in incorrect sample attribution.
>
> | Method                                  | Avg. highest conf. (0-25%) (↑) | Avg. highest conf. (0-50%) (↑) | Avg. highest conf. (0-75%) (↑) | Avg. highest conf. (0-100%) (↑) | Insertion ($\uparrow$) |
> | --------------------------------------- | :----------------------------: | :----------------------------: | :----------------------------: | :-----------------------------: | :--------------------: |
> | HSIC-Attribution (Novello et al., 2022) |             0.1709             |             0.2091             |             0.2250             |             0.2493              |         0.1446         |
> | Patch 5$\times$5                        |           **0.2597**           |             0.3933             |             0.4389             |             0.4515              |       **0.1708**       |
> | Patch 6$\times$6                        |             0.2372             |             0.4025             |             0.4555             |             0.4720              |         0.1538         |
> | Patch 7$\times$7                        |             0.2430             |           **0.4289**           |             0.4819             |             0.4985              |         0.1621         |
> | Patch 8$\times$8                        |             0.1903             |             0.4005             |             0.4740             |             0.5043              |         0.1584         |
> | Patch 10$\times$10                      |             0.2020             |             0.4065             |             0.4908             |             0.5237              |         0.1519         |
> | Patch 12$\times$12                      |             0.1667             |             0.3816             |           **0.4987**           |           **0.5468**            |         0.1247         |
>
> ----
>
> **W4:** In the introduction, “and a fine-grained dataset CUB-200-2011 (Welinder et al., 2010) datasets” -> “and a fine-grained dataset CUB-200-2011 (Welinder et al., 2010)”
>
> **AW4:** Thanks for your kind reminder, we have corrected it in the revised manuscript.
>
> ----
>
> **W5:** In Algorithm 1, the input k is not used, please check carefully.
>
> **AW5:** Thanks for your kind reminder, $n$ should be corrected to $k$, we have corrected it in the revised manuscript.

---

> > ### Comment · Reviewer_2VHY · 2023-11-21
> > **Raising the score to 8**
> >
> > Thanks for your response. I have carefully read the response and other reviews. I think the authors have addressed my concerns. Hence, I raise my score to 8.

---

> > > ### Author Response · Authors · 2023-11-21
> > > **Thanks to Reviewer 2VHY**
> > >
> > > We express our gratitude for the time and effort you have dedicated as a reviewer for ICLR 2024. Your valuable suggestions and the suggestion of relevant experiments have significantly contributed to the improvement of our article. We deeply appreciate your recognition and support of our work.

---

### Official Review · Reviewer_gx4h · 2023-10-30

**Soundness:** 3 good
**Presentation:** 3 good
**Contribution:** 4 excellent
**Rating:** 8
**Confidence:** 4

**Summary:**

This article transforms the image attribution problem into a submodular subset selection problem and determines the importance of divided regions through a greedy search algorithm. The author designed the submodular function from four aspects, hoping to use fewer areas to obtain a higher interpretable area. The author verified the effectiveness of this method on two tasks: face recognition and fine-grained recognition. Experimental results show that this method can achieve better attribution effects and can better debug incorrectly predicted samples.

**Strengths:**

- This paper reformulates the attribution problem as a submodular subset selection problem that achieves higher interpretability with fewer fine-grained regions.

- The proposed method enables more accurate attribution and can help find the reasons for the model to produce incorrect prediction results.

- It is meaningful to verify this interpretable method on face recognition and fine-grained recognition tasks, because these tasks are closer to practical applications.

- The authors provide some theoretical guarantees.

**Weaknesses:**

- In Algorithm 1, I didn't see the use of variable k. Should n of line 3 be k?

- In Table 1, why are some results of LIME and Kernel Shap not reported? Is it because these attribution algorithms have limitations on the CUB data set? Hope the author can explain it.

- It would be better if the authors could discuss the limitations of this method.

- Can the author further state whether the proposed method is white-box based or black-box based (assuming that the calculation of a priori saliency map is not considered)?

**Questions:**

Listed in the weakness of the paper.

**Details Of Ethics Concerns:**

No ethics concern.

---

> ### Author Response · Authors · 2023-11-20
> **Official Response to Reviewer gx4h**
>
> Thanks for your encouraging evaluations and constructive comments. We provide a detailed response, supplemented with additional experiments, to address your questions and concerns.
>
> ----
>
> **W1:** In Algorithm 1, I didn't see the use of variable k. Should n of line 3 be k?
>
> **AW1:** Thanks for your kind reminder, this was a math typo we made, the $n$ in the algorithm should be $k$, and we have corrected it in the revised paper.
>
> ----
>
> **W2:** In Table 1, why are some results of LIME and Kernel Shap not reported? Is it because these attribution algorithms have limitations on the CUB data set? Hope the author can explain it.
>
> Thanks for your kind reminder. This limitation arises when using the Xplique library as both LIME and Kernel Shap methods are unable to calculate the saliency map for the ResNet-101 network trained on the CUB-200-2011 data set. This issue may be inherent to the Xplique library, leading us not to report relevant experiments.
>
> We have repeated this experiment based on other libraries, using the same network. The table below demonstrates the effectiveness of our method when applied to LIME and Kernel Shap on the CUB-200-2011 dataset. We have also included these experimental results in the revised manuscript.
>
> | Method                | Deletion ($\downarrow$) | Insertion ($\uparrow$) |
> | --------------------- | :---------------------: | :--------------------: |
> | LIME                  |         0.1070          |         0.6812         |
> | LIME (w/ ours)        |       **0.0941**        |       **0.6994**       |
> | Kernel Shap           |         0.1016          |         0.6763         |
> | Kernel Shap (w/ ours) |       **0.0951**        |       **0.6920**       |
>
> ----
>
> **W3:** It would be better if the authors could discuss the limitations of this method.
>
> **AW3:** Thanks for your helpful suggestion. We have added a section discussing the limitations of this method in the Appendix.
>
> The main limitation of our method is the computation time depending on the sub-region division approach. Specifically, we first divide the image into different subregions, and then we use the greed algorithm to search for the important subset regions. To accelerate the search process, we introduce the prior maps, which are computed based on the existing attribution methods. However, we observe that the performance of our attribution method is based on the scale of subregions, where the smaller regions would achieve much better accuracy as shown in the experiments. There exists a trade-off between the accuracy and the computation time. In the future, we will explore a better optimization strategy to solve this limitation.
>
> We performed experiments on the attribution of incorrectly predicted samples and examined the effects of varying set sizes on the experimental results, using the network ResNet-101. As shown in the table below, a larger set size $m$ results in an increase in the average highest confidence within the search region. However, it's important to note that this improvement also leads to a rise in the computational time cost.
>
> |       Method       | Set size $m$ | Avg. highest conf. (0-100%) (↑) | Time consumption per image (s) |
> | :----------------: | :----------: | :-----------------------------: | :----------------------------: |
> |  Patch 5$\times$5  |      25      |             0.4515              |              12.3              |
> |  Patch 6$\times$6  |      36      |             0.4720              |              16.8              |
> |  Patch 7$\times$7  |      49      |             0.4985              |              34.2              |
> |  Patch 8$\times$8  |      64      |             0.5043              |              61.3              |
> | Patch 10$\times$10 |     100      |             0.5237              |             156.7              |
> | Patch 12$\times$12 |     144      |           **0.5468**            |             345.1              |
>
> ----
>
> **W4:** Can the author further state whether the proposed method is white-box based or black-box based (assuming that the calculation of a priori saliency map is not considered)?
>
> **AW4:** Thanks for your helpful suggestion. Without considering the calculation of the prior saliency map, our proposed attribution method belongs to the black-box-based explanation method. The black-box interpretation is defined as that these models make predictions based on input data, but the decision-making process and reasoning behind the predictions are not transparent to the user. In this paper, our method is only using the output of the specific model without requiring the other information. Thus, our proposed attribution method is the black-box-based approach.

---

> > ### Comment · Reviewer_gx4h · 2023-11-21
> >
> > Thank you for your response. It helps clear up my concerns, especifally for the detailed discussion on the limitations of this method.  I have also carefully read comments from other reviewers. Considering the novel solution based on a submodular subset selection problem,  the SoTA performance, and the helpful response, I decide to increase my score to 8.

---

> > > ### Author Response · Authors · 2023-11-21
> > > **Thanks to Reviewer gx4h**
> > >
> > > We extend our sincere thanks for your valuable time and effort in serving as a reviewer for ICLR 2024. We greatly appreciate your constructive feedback and the recognition you have accorded to our work.

---

### Official Review · Reviewer_2zii · 2023-11-01

**Soundness:** 3 good
**Presentation:** 2 fair
**Contribution:** 2 fair
**Rating:** 6
**Confidence:** 2

**Summary:**

The authors of this paper propose a novel explainability method that restates the image attribution problem as a submodular subset selection problem. They suggest they can achieve better interpretability results with local regions. Moreover, they aim to obtain higher scores of interpretability with fewer regions.

**Strengths:**

- It is impressive that their method achieves a 81.0% gain in the average highest confidence score for incorrectly predicted samples.
- Treating the interpretable region identification problem as a submodular subset selection problem is a novel and interesting idea.
- Their method has the ability to find the reasons that causes the prediction error for incorrectly predicted images.
- Adding the ablation study at the end is a good idea.

**Weaknesses:**

Some suggestions for minor improvements:
- The phrase "... at the level of theoretical aspects."at the introduction sounds a bit too wordy. It can be expressed more concisely.
- This sentence in the introduction "Image attribution algorithm is a typical interpretable method, which produces saliency
maps that explain how important image regions are to model decisions." can be better phrased, in my opinion, as "... that explain which image regions are more important to model decisions."
- I don't think fine-grainedness (page 2) is an actual word. Fine-graininess may be an alternative but I am not sure.
- On page 2, the word "...datasets." at the end of the first sentence of the second paragraph needs to be omitted.
- Contrary to what has been said on the introductory sentence of the White-Box Attribution method paragraph, I don't think there is a THE image attribution algorithm. I advise the authors to state either the name of the specific algorithm they are mentioning or use the plural.

**Questions:**

- How do the image attribution algorithms relate to attention in neural networks in general?
- What is the meaning of fine-grained interpretation regions?
- What does the "validity" of a submodular function mean?
- Is there an explanation on why the blue curve on the left of Figure 1 dips to almost 0 around 0.8?
- You claim your method can find fewer regions that make the model predictions more confident but I am seeing more highlighted regions in your setting. Am I missing something or misreading the images? Can you elaborate more on what I should be seeing?
- How does your method compare to the SOTA HSIC-Attribution method in terms of algorithmic complexity and time?

---

> ### Author Response · Authors · 2023-11-20
> **Official Response to Reviewer 2zii (1/2)**
>
> Thanks for your encouraging evaluations and constructive comments. We provide a detailed response, to address your questions and concerns.
>
> ----
>
> **W1:** The phrase "... at the level of theoretical aspects."at the introduction sounds a bit too wordy. It can be expressed more concisely.
>
> **AW1:** Thanks for your kind reminder. To make it more concise, we revised this sentence as follows:
>
> > Furthermore, our analysis at the theoretical level confirms that the proposed function is indeed submodular.
>
> ----
>
> **W2:** This sentence in the introduction "Image attribution algorithm is a typical interpretable method, which produces saliency maps that explain how important image regions are to model decisions." can be better phrased, in my opinion, as "... that explain which image regions are more important to model decisions."
>
> **AW2:** Thanks for your kind reminder. We revised this sentence as follows:
>
> > Image attribution algorithm is a typical interpretable method, which produces saliency maps that explain which image regions are more important to model decisions.
>
> ----
>
> **W3:** I don't think fine-grainedness (page 2) is an actual word. Fine-graininess may be an alternative but I am not sure.
>
> **AW3:** Thanks for your kind reminder, "fine-grainedness" should be corrected to "fine-graininess", we have corrected it in the revised manuscript.
>
> ----
>
> **W4:** On page 2, the word "...datasets." at the end of the first sentence of the second paragraph needs to be omitted.
>
> **AW4:** Thanks for your kind reminder, we have removed it in the revised manuscript.
>
> ----
>
> **W5:** Contrary to what has been said on the introductory sentence of the White-Box Attribution method paragraph, I don't think there is a THE image attribution algorithm. I advise the authors to state either the name of the specific algorithm they are mentioning or use the plural.
>
> **AW5:** Thanks for your kind reminder, we have revised the description of this part as follows:
>
> > Image attribution algorithms are designed to ascertain the importance of different input regions within an image with respect to the decision-making process of the model.

---

> ### Author Response · Authors · 2023-11-20
> **Official Response to Reviewer 2zii (2/2)**
>
> **Q1:** How do the image attribution algorithms relate to attention in neural networks in general?
>
> **AQ1:** Image attribution algorithms refer to explaining a model's behavior, i.e., attributing its decision to pivotal features. The higher attribution score indicates the more importance of the decision. Attention in neural networks is the visual similarity between different feature representations, where the higher value only indicates the higher feature similarity. Maybe the attention could be used for image attribution, which we will explore for future work.
>
> ----
>
> **Q2:** What is the meaning of fine-grained interpretation regions?
>
> **AQ2:** In this paper, we use the fine-grained interpretation regions to represent the density of the divided image regions, e.g., $7 \times 7$ and $10 \times 10$. The HSIC-Attribution (Novello et al., 2022) method typically operates at the $7 \times 7$ patch level, treating each patch as equally significant. However, when the density of attributed patches increases, say to $10 \times 10$, its effectiveness in attribution tends to diminish, indicating a limitation. Consequently, in our method, we aim to enhance the density of the divided image regions. This strategy is intended to yield more detailed interpretations when attribution while still maintaining the efficacy of the attribution results.
>
> [Reference] Paul Novello, Thomas Fel, and David Vigouroux. Making sense of dependence: Efficient black-box explanations using dependence measure. In Annual Conference on Neural Information Processing Systems (NeurIPS), pp. 4344–4357, 2022.
>
> ----
>
> **Q3:** What does the "validity" of a submodular function mean?
>
> **AQ3:** Thank you for your question. We aim to express that the proposed function $\mathcal{F}(\cdot)$ is a submodular function, characterized by its nonnegative monotonicity and compliance with Equation 1. For greater clarity, we have revised the sentence in the abstract as follows:
>
> > Moreover, our theoretical analysis substantiates that the proposed function is in fact submodular.
>
>  and we also have revised the sentence in the introduction as follows:
>
> > Furthermore, our analysis at the theoretical level confirms that the proposed function is indeed submodular.
>
> ----
>
> **Q4:** Is there an explanation on why the blue curve on the left of Figure 1 dips to almost 0 around 0.8?
>
> **AQ4:** Thanks for your question. The blue curve denotes the prediction score for the specific category, which is computed based on the discovered subset regions. Since the visual similarity between different bird categories is high, some categories are easily confused based on the partial features. In Fig.1,  with the growth of the discovered regions, we may introduce some features for the other category, and thus the prediction score drops for the zeros. We also provide more visualization results in the Appendix.
>
> ----
>
> **Q5:** You claim your method can find fewer regions that make the model predictions more confident but I am seeing more highlighted regions in your setting. Am I missing something or misreading the images? Can you elaborate more on what I should be seeing?
>
> **AQ5:** Thanks for your kind reminder. The advantage of our proposed attribution method is to discover the discriminative regions for the prediction. If we discovered the correct subregions, the prediction confidence would be higher than using the entire image, especially for fine-grained object recognition.
>
> For example, as illustrated in Figure 1, when the entire image is fed into the recognition model, the confidence level for the correctly predicted category is approximately 0.7. However, when utilizing only 25% of the image area, as identified by our method, the model's confidence level for the correct category exceeds 0.7. This indicates that our method enables the model to make predictions with higher confidence using fewer regions.
>
> ----
>
> **Q6:** How does your method compare to the SOTA HSIC-Attribution method in terms of algorithmic complexity and time?
>
> **AQ6:** Our method requires the initial computation of a saliency map to facilitate the division of sub-regions. When employing HSIC-Attribution as the baseline, this implies that both the time and space complexity of our approach depend on those of HSIC-Attribution, while also taking into account the time and space complexity associated with the greedy search component. Consequently, achieving superior attribution results with our method may require a higher time complexity compared to HSIC-Attribution.

---

### Official Review · Reviewer_kDMz · 2023-11-09

**Soundness:** 3 good
**Presentation:** 2 fair
**Contribution:** 3 good
**Rating:** 8
**Confidence:** 5

**Summary:**

The paper points out two main issues with current State-of-the-Art (SoTA) image attribution methods: they ignore the impact of local, fine-grained attribution regions which may lead to incorrect explanations, and they may struggle to map the region/cause of a prediction error to image samples.

To address these problems, the authors propose a new method based on submodular functions that divides an image into smaller regions and then selects the most informative ones (subset selection) to explain the model's decision-making process. They also employ a regional search to expand on the search regions.
The paper also introduces a novel submodular function to provide clearer and more detailed explanations, especially for incorrect predictions based on four major clues - confidence, effectiveness, consistency and collaboration scores.

The effectiveness of this new method is supported by experiments on facial recognition and fine-grained image recognition datasets, where it is shown to perform better than the current SoTA methods.

**Strengths:**

1. The paper highlights the importance of fine-grained, local regions for image attribution alongside causation of erroneous predictions to image features.
2. The novel submodular function proposed in the paper has been demonstrated to outperform several State-of-the-Art (SoTA) approaches.
3. The paper also introduce several interpretability clues such as confidence $s_{conf}$, effectiveness $s_{eff.}$, consistency $s_{cons.}$ and collaboration $s_{colla.}$ to evaluate the significance of the selected subsets. These additions effectively demonstrate better interpretability and are well supported by theoretical and empirical results.

**Weaknesses:**

1. The idea of decomposing an input image $\mathbf{I}$ into regions has been studied for several vision tasks like self-supervision (Noroozi et al., 2016 etc.), object detection (Redmon and Farhadi, 2018) etc. These should be cited and the differences should be called out.
2. The use of saliency maps $A$ in sub-region division is unclear. The paper should highlight how saliency maps are used to evaluate patch importance.
3. Most of the scoring functions like  $s_{eff.}$, $s_{cons.}$ etc.  rely on cosine similarity or distance metrics which have been studied extensively in literature (Deng et al., 2018, Wang et al., 2018 etc.) but have not been cited in the paper.
4. The paper lacks the explanation regarding how individual scores contribute to achieving their respective objectives. For example, $s_{colla.}$ employs a cosine distance metric between the semantic feature vector of the target class $f_s$ and features extracted from the residual regions of the original image when the selected subset of regions $S$ is removed. To the best of my knowledge, maximizing this metric ensures that the collective impact of the selected region is sufficient to generate explainable representations.
5. The proposed greedy search algorithm (section 4.3) has been studied for subset selection tasks (Wei et al., 2015) and is therefore prior art.
6. The paper misses a critical reference in submodular optimization (Fujishige, 2005) and should include it in the related work.
7. The experiments should include ablations on $k$ which is the number of sub-regions selected from $V$.

**Questions:**

Most of the suggestions have been explained in detail in the weakness section. Additionally, some additional suggestions are listed below:
1. Section 4.1 which highlights the Sub-Region Division should be presented as an algorithm for better clarity.
2. It would be great to label the Input Image as $\mathbf{I}$ and saliency / attribution map as $\mathbf{A}$ in figure 1 for better clarity.
3. It is unclear as to what $M$ signifies in the problem definition, which should be clarified in sections 3 and 4.2.
4. Lemmas 1 and 2 alongside equations 10 and 11 are already discussed in the problem definition in section 3, thus should be removed with referencing.
5. Theorem 1 is prior art (Nemhauser et al., 1978) and should just be cited.
6. Although optional, it would be good to have a set of notations as this paper encapsulates multiple domains in Machine Learning.
7. Experiments in section 5 indicate that the proposed method demonstrates significant improvements over existing methods. This shows the generalizability of the approach irrespective of the underlying model, which I believe can be highlighted for better impact.

---

> ### Author Response · Authors · 2023-11-20
> **Official Response to Reviewer kDMz (1/8)**
>
> Thanks for your encouraging evaluations and constructive comments. We provide a detailed response, supplemented with additional experiments, to address your questions and concerns.
>
> ----
>
> **W1:** The idea of decomposing an input image $\mathbf{I}$ into regions has been studied for several vision tasks like self-supervision (Noroozi et al., 2016 etc.), object detection (Redmon and Farhadi, 2018) etc. These should be cited and the differences should be called out.
>
> **AW1:** Thanks for your helpful suggestion. Traditional methods (Noroozi & Favaro, 2016; Redmon & Farhadi, 2018) typically divide images into regular patch areas, neglecting the semantic information inherent in different regions. In contrast, our method employs a sub-region division strategy that is informed and guided by an a priori saliency map. Therefore, our method could divide the regions based on their contribution to the prediction. We have shown the attribution performances based on different region divided methods as shown in Table 4.  From the experiment, we observe that the attribution results after introducing a priori saliency map to guide sub-region division are better than dividing image patches regularly.
>
> We have cited related work and highlighted the differences in the revised manuscript.
>
> Table 4: Impact on whether to use a priori attribution map.
>
> | Method               | Set size $m$ | Celeb-A Deletion ($\downarrow$) | Celeb-A Insertion ($\uparrow$) | CUB Deletion ($\downarrow$) | CUB Insertion ($\uparrow$) |
> | -------------------- | :----------: | :-----------------------------: | :----------------------------: | :-------------------------: | :------------------------: |
> | Patch $7 \times 7$   |      49      |             0.1493              |             0.5642             |           0.1061            |           0.6903           |
> | Patch $10 \times 10$ |     100      |             0.1365              |             0.5459             |           0.1024            |           0.6159           |
> | Patch $14 \times 14$ |     196      |             0.1284              |             0.5562             |           0.0853            |           0.5805           |
> | +HSIC-Attribution    |      25      |           **0.1054**            |           **0.5752**           |         **0.0613**          |         **0.7262**         |
>
> [Reference A] Mehdi Noroozi and Paolo Favaro. Unsupervised learning of visual representations by solving jigsaw puzzles. In European Conference on Computer Vision (ECCV), pp. 69–84. Springer, 2016.
>
> [Reference B] Joseph Redmon and Ali Farhadi. Yolov3: An incremental improvement. arXiv preprint arXiv:1804.02767, 2018.

---

> ### Author Response · Authors · 2023-11-20
> **Official Response to Reviewer kDMz (2/8)**
>
> **W2:** The use of saliency maps $\mathcal{A}$ in sub-region division is unclear. The paper should highlight how saliency maps are used to evaluate patch importance.
>
> **AW2:** Thanks for your kind reminder. In detail, we initially divide the image into $N \times N$ patch regions. Subsequently, an existing image attribution algorithm is applied to compute the saliency map $\mathcal{A}\in \mathbb{R}^{w \times h}$ for a corresponding class of $\mathbf{I}$. Following this, we resize $\mathcal{A}$ to a $N \times N$ dimension, where its values denote the importance of each patch. Based on the determined importance of each patch, $d$ patches are sequentially allocated to each sub-region $\mathbf{I}^M$, while the remaining patch regions are masked with $\mathbf{0}$, where $d = N \times N / m$. This process finally forms the element set
> $V= \{\mathbf{I_{1}}^{M},\mathbf{I_{2}}^{M},\cdots,\mathbf{I_{m}}^{M}\}$, satisfying the condition $\mathbf{I} = \sum_{i=1}^{m}\mathbf{I}^{M}_{i}$. The detailed calculation process is outlined in Algorithm 1.
>
> > **Algorithm 1:** A greedy search based algorithm for interpretable region discovery
> >
> > ---
> >
> > **Input:** Image $\mathbf{I} \in \mathbb{R}^{w \times h \times 3}$, number of divided patches $N \times N$, a prior saliency map $\mathcal{A}\in \mathbb{R}^{w \times h \times 3}$ of $\mathbf{I}$, number of image division sub-regions $m$, maximum number of constituent regions $k$.
> >
> > **Output:** A set $S \subseteq V$, where $\left | S \right |  \le k$.
> >
> > $V \gets \varnothing$  &emsp;&emsp;&emsp;&emsp;   /\* Initiate the operation of sub-region division \*/
> >
> > $\mathcal{A} \gets \text{resize}\left(\mathcal{A}, \text{newRows}=N, \text{newCols}=N\right)$
> >
> > $d = N \times N / m$
> >
> > **for** $l=1$ **to** $m$ **do**:
> >
> > ​&emsp;$\mathbf{I}^{M}_{l} = \mathbf{I}$
> >
> > ​&emsp;**for** $i=1$ **to** $N$ **do**:
> >
> > ​&emsp;&emsp;**for** $j=1$ **to** $N$ **do**:
> >
> > ​&emsp;&emsp;&emsp;$I_r=\text{rank}\left(\mathcal{A}, i, j \right)$	/\* Index of $\mathcal{A}_{i,j}$, ordered descendingly \*/
> >
> > &emsp;&emsp;&emsp;**if** $I_r < (d-1) \times l$ and $I_r > d \times l$ **then**:
> >
> > &emsp;&emsp;&emsp;&emsp;$\mathbf{I}^{M}_{l}\left[(I_r-1)\times w/N+1 : I_r\times w/N, (I_r-1)\times h/N+1 : I_r\times h/N \right] = \mathbf{0}$
> >
> > ​&emsp;&emsp;**end**
> >
> > ​&emsp;**end**
> >
> > ​&emsp;$V \gets V \cup \{\mathbf{I}^{M}_{l} \}$
> >
> > **end**
> >
> > ...

---

> ### Author Response · Authors · 2023-11-20
> **Official Response to Reviewer kDMz (3/8)**
>
> **W3:** Most of the scoring functions like $s_{eff.}$., $s_{cons.}$. etc. rely on cosine similarity or distance metrics which have been studied extensively in literature (Deng et al., 2018, Wang et al., 2018 etc.) but have not been cited in the paper.
>
> **AW3:** Thanks for your helpful suggestion. Traditional distance measurement methods (Deng et al., 2018, Wang et al., 2018) are tailored to maximize the decision margins between classes during model training, involving operations like feature scaling and increasing angle margins. In contrast, our approach focuses solely on calculating the relative distance between features, for which we utilize the general cosine distance. We have revised it in the manuscript and cited relevant references.
>
> [Reference C] Hao Wang, Yitong Wang, Zheng Zhou, Xing Ji, Dihong Gong, Jingchao Zhou, Zhifeng Li, and Wei Liu. Cosface: Large margin cosine loss for deep face recognition. In IEEE Conference on Computer Vision and Pattern Recognition (CVPR), pp. 5265–5274, 2018.
>
> [Reference D] Jiankang Deng, Jia Guo, Niannan Xue, and Stefanos Zafeiriou. Arcface: Additive angular margin loss for deep face recognition. In IEEE Conference on Computer Vision and Pattern Recognition (CVPR), pp. 4690–4699, 2019.

---

> ### Author Response · Authors · 2023-11-20
> **Official Response to Reviewer kDMz (4/8)**
>
> **W4:** The paper lacks the explanation regarding how individual scores contribute to achieving their respective objectives. For example, $s_{colla.}$. employs a cosine distance metric between the semantic feature vector of the target class $\boldsymbol{f}_{s}$ and features extracted from the residual regions of the original image when the selected subset of regions $S$ is removed. To the best of my knowledge, maximizing this metric ensures that the collective impact of the selected region is sufficient to generate explainable representations.
>
> **AW4:** Thanks for your helpful suggestion. We explain the objectives of these scores in more detail in Section 4.2.
>
> The confidence score $s_{\mathrm{conf.}}$ is designed to distinguish regions from out-of-distribution, ensuring alignment with the InD. Thus, we introduce Evidential Deep Learning to metric the uncertainty of an input sample and compute the confidence score. By incorporating the $s_{\mathrm{conf.}}$, we can ensure that the selected regions align closely with the In-Distribution (InD).
>
> The effectiveness score $s_{\mathrm{eff.}}$ is designed to increase the diversity and improving the overall quality of region selection. Thus, we introduce cosine distance to metric the similarity between different sub-regions. If in the selected subset, the distance between the two nearest sub-regions may be large, indicating that the diversity of its special regions is high. By incorporating the $s_{\mathrm{eff.}}$, we aim to limit the selection of regions with similar semantic representations.
>
> The consistency score $s_{\mathrm{cons.}}$ is designed to ensure a precise selection that aligns closely with our specific semantic goals. Thus, we introduce cosine distance to metric the similarity between the selected region with target semantic feature. By incorporating the $s_{\mathrm{cons.}}$, our method targets regions that reinforce the desired semantic response.
>
> The collaboration score $s_{\mathrm{colla.}}$ is designed to find regions of high collective effect. Such a metric is particularly valuable in the initial stages of the search, highlighting regions essential for sustaining the model's accuracy and reliability. Thus, we introduce cosine distance to metric the similarity between the unselected region with target semantic feature. As a result, the similarity level decreases rapidly, and the selected area has a very high collective effect. By incorporating the $s_{\mathrm{colla.}}$, our method pinpoints regions whose exclusion markedly affects the model's predictive confidence.
>
> In the experimental part, we conducted ablation experiments on various combinations of these scores. As shown in Table 3, we observed that using a single score function within the submodular function imposes limitations on attribution faithfulness. The experimental results illustrate that removing any of the four score functions leads to deteriorated Deletion and Insertion scores, thereby confirming the validity of these score functions.
>
> Table 3: Ablation study on components of different score functions of submodular function on the Celeb-A, and CUB-200-2011 validation sets.
>
> | Submodular Function  |                     |                      |                       |         Celeb-A         |        Celeb-A         |      Cub-200-2011       |      Cub-200-2011      |
> | :------------------: | :-----------------: | :------------------: | :-------------------: | :---------------------: | :--------------------: | :---------------------: | :--------------------: |
> | $s_{\mathrm{conf.}}$ | $s_{\mathrm{eff.}}$ | $s_{\mathrm{cons.}}$ | $s_{\mathrm{colla.}}$ | Deletion ($\downarrow$) | Insertion ($\uparrow$) | Deletion ($\downarrow$) | Insertion ($\uparrow$) |
> |                      |       $\surd$       |       $\surd$        |        $\surd$        |         0.1074          |         0.5735         |         0.0632          |         0.7169         |
> |       $\surd$        |                     |       $\surd$        |        $\surd$        |         0.1993          |         0.2616         |         0.0623          |         0.7227         |
> |       $\surd$        |       $\surd$       |                      |        $\surd$        |         0.1067          |         0.5712         |         0.0651          |         0.6753         |
> |       $\surd$        |       $\surd$       |       $\surd$        |                       |         0.1088          |         0.5750         |         0.0811          |         0.7090         |
> |       $\surd$        |       $\surd$       |       $\surd$        |        $\surd$        |       **0.1054**        |       **0.5752**       |       **0.0613**        |       **0.7262**       |
>
> continue down

---

> ### Author Response · Authors · 2023-11-20
> **Official Response to Reviewer kDMz (5/8)**
>
> Continue From Above
>
> Similarly, within the experiment of incorrect sample attribution (refer to Table 10), it shows that the removal of any score function, regardless of the imputation algorithm it is based on, results in a decrease in both the average highest confidence and Insertion AUC score.
>
> Table 10: Ablation study on submodular function score components for incorrectly predicted samples in the CUB-200-2011 dataset.
>
> | Method                     | $s_{cons.}$ | $s_{colla.}$ | Avg. highest conf. (0-25%) (↑) | Avg. highest conf. (0-50%) (↑) | Avg. highest conf. (0-75%) (↑) | Avg. highest conf. (0-100%) (↑) | Insertion ($\uparrow$) |
> | -------------------------- | :---------: | :----------: | :----------------------------: | :----------------------------: | :----------------------------: | :-----------------------------: | :--------------------: |
> | Grad-CAM++ (w/ ours)       |             |   $\surd$    |             0.0821             |             0.1547             |             0.1923             |             0.2303              |         0.1122         |
> | Grad-CAM++ (w/ ours)       |   $\surd$   |              |             0.1654             |             0.2888             |             0.3338             |             0.3611              |         0.1452         |
> | Grad-CAM++ (w/ ours)       |   $\surd$   |   $\surd$    |           **0.2424**           |           **0.3575**           |           **0.3934**           |           **0.4193**            |       **0.1672**       |
> | Score-CAM (w/ ours)        |             |   $\surd$    |             0.0742             |             0.1348             |             0.1835             |             0.2237              |         0.1072         |
> | Score-CAM (w/ ours)        |   $\surd$   |              |             0.1383             |             0.2547             |             0.3131             |             0.3402              |         0.1306         |
> | Score-CAM (w/ ours)        |   $\surd$   |   $\surd$    |           **0.2491**           |           **0.3395**           |           **0.3796**           |           **0.4082**            |       **0.1622**       |
> | HSIC-Attribution (w/ ours) |             |   $\surd$    |             0.1054             |             0.1803             |             0.2177             |             0.2600              |         0.1288         |
> | HSIC-Attribution (w/ ours) |   $\surd$   |              |             0.2394             |             0.3479             |             0.3940             |             0.4220              |         0.1645         |
> | HSIC-Attribution (w/ ours) |   $\surd$   |   $\surd$    |           **0.2430**           |           **0.3519**           |           **0.3984**           |           **0.4513**            |       **0.1772**       |

---

> ### Author Response · Authors · 2023-11-20
> **Official Response to Reviewer kDMz (6/8)**
>
> **W5:** The proposed greedy search algorithm (section 4.3) has been studied for subset selection tasks (Wei et al., 2015) and is therefore prior art.
>
> **AW5:** Thanks for your kind reminder. Indeed, the greedy search algorithm we employ is well-established in existing literature. Wei et al. select sample sets in the dataset through a greedy search algorithm, while our method mainly selects image regions. Our tasks and the defined submodular functions are completely different. We have made sure to cite related works, specifically those by Mirzasoleiman et al., 2015, and Wei et al., 2015, in our paper.
>
> [Reference E] Baharan Mirzasoleiman, Ashwinkumar Badanidiyuru, Amin Karbasi, Jan Vondrak, and Andreas Krause. Lazier than lazy greedy. In AAAI Conference on Artificial Intelligence (AAAI), pp. 1812– 1818, 2015.
>
> [Reference F] Kai Wei, Rishabh Iyer, and Jeff Bilmes. Submodularity in data subset selection and active learning. In International Conference on Machine Learning (ICML), pp. 1954–1963. PMLR, 2015.
>
> ----
>
> **W6:** The paper misses a critical reference in submodular optimization (Fujishige, 2005) and should include it in the related work.
>
> **AW6:** Thanks for your kind reminder, we have cited the references you suggested in the related work:
>
> > Submodular optimization (Fujishige, 2005) has been successfully studied in multiple application scenario.
>
> [Reference G] Satoru Fujishige. Submodular functions and optimization. Elsevier, 2005.
>
> ----
>
> **W7:** The experiments should include ablations on $k$ which is the number of sub-regions selected from $V$.
>
> **AW7:** Thanks for your kind reminder. In image attribution tasks, it is essential to sort all sub-regions in the set $V$ since each sub-region contributes to the prediction. As highlighted in our experimental settings, to effectively validate the attribution effect, the value of $k$ will be specifically set equal to $m$ (the size of the set $V$).
>
> In Table 2, we presented the average highest confidence evaluation metric across various search ranges. For this, we established four search region ranges: 0-25%, 0-50%, 0-75%, and 0-100%, corresponding to setting $k$ to $0.25m$, $0.5m$, $0.75m$, and $m$, respectively. The experimental results indicate that a larger $k$ value generally yields a higher average highest confidence, although the increase tends to plateau. This is because the highest confidence does not decrease as the search range becomes larger.
>
> Table 2. Evaluation of discovering the cause of incorrect predictions.
>
> | Method                     | Avg. highest conf. (0-25%) (↑) | Avg. highest conf. (0-50%) (↑) | Avg. highest conf. (0-75%) (↑) | Avg. highest conf. (0-100%) (↑) | Insertion ($\uparrow$) |
> | :------------------------- | :----------------------------: | :----------------------------: | :----------------------------: | :-----------------------------: | :--------------------: |
> | Grad-CAM++                 |             0.1988             |             0.2447             |             0.2544             |             0.2647              |         0.1094         |
> | Grad-CAM++ (w/ ours)       |           **0.2424**           |           **0.3575**           |           **0.3934**           |           **0.4193**            |       **0.1672**       |
> | Score-CAM                  |             0.1896             |             0.2323             |             0.2449             |             0.2510              |         0.1073         |
> | Score-CAM (w/ ours)        |           **0.2491**           |           **0.3395**           |           **0.3796**           |           **0.4082**            |       **0.1622**       |
> | HSIC-Attribution           |             0.1709             |             0.2091             |             0.2250             |             0.2493              |         0.1446         |
> | HSIC-Attribution (w/ ours) |           **0.2430**           |           **0.3519**           |           **0.3984**           |           **0.4513**            |       **0.1772**       |

---

> ### Author Response · Authors · 2023-11-20
> **Official Response to Reviewer kDMz (7/8)**
>
> **Q1:** Section 4.1 which highlights the Sub-Region Division should be presented as an algorithm for better clarity.
>
> **AQ1:** Thanks for your helpful suggestion. We have added the detailed calculation process of the Sub-region Division in Algorithm 1:
>
> > **Algorithm 1:** A greedy search based algorithm for interpretable region discovery
> >
> > ---
> >
> > **Input:** Image $\mathbf{I} \in \mathbb{R}^{w \times h \times 3}$, number of divided patches $N \times N$, a prior saliency map $\mathcal{A}\in \mathbb{R}^{w \times h \times 3}$ of $\mathbf{I}$, number of image division sub-regions $m$, maximum number of constituent regions $k$.
> >
> > **Output:** A set $S \subseteq V$, where $\left | S \right |  \le k$.
> >
> > $V \gets \varnothing$  &emsp;&emsp;&emsp;&emsp;   /\* Initiate the operation of sub-region division \*/
> >
> > $\mathcal{A} \gets \text{resize}\left(\mathcal{A}, \text{newRows}=N, \text{newCols}=N\right)$
> >
> > $d = N \times N / m$
> >
> > **for** $l=1$ **to** $m$ **do**:
> >
> > ​&emsp;$\mathbf{I}^{M}_{l} = \mathbf{I}$
> >
> > ​&emsp;**for** $i=1$ **to** $N$ **do**:
> >
> > ​&emsp;&emsp;**for** $j=1$ **to** $N$ **do**:
> >
> > ​&emsp;&emsp;&emsp;$I_r=\text{rank}\left(\mathcal{A}, i, j \right)$	/\* Index of $\mathcal{A}_{i,j}$, ordered descendingly \*/
> >
> > &emsp;&emsp;&emsp;**if** $I_r < (d-1) \times l$ and $I_r > d \times l$ **then**:
> >
> > &emsp;&emsp;&emsp;&emsp;$\mathbf{I}^{M}_{l}\left[(I_r-1)\times w/N+1 : I_r\times w/N, (I_r-1)\times h/N+1 : I_r\times h/N \right] = \mathbf{0}$
> >
> > ​&emsp;&emsp;**end**
> >
> > ​&emsp;**end**
> >
> > ​&emsp;$V \gets V \cup \{\mathbf{I}^{M}_{l} \}$
> >
> > **end**
> >
> > ...

---

> ### Author Response · Authors · 2023-11-20
> **Official Response to Reviewer kDMz (8/8)**
>
> **Q2:** It would be great to label the Input Image as $\mathbf{I}$ and saliency / attribution map as $\mathcal{A}$ in figure 1 for better clarity.
>
> **AQ2:** Thanks for your helpful suggestion. We added input images and saliency maps along with their mathematical symbols as input in Figure 1. We have also updated Figure 2 and marked the above symbols in the figure.
>
> ----
>
> **Q3:** It is unclear as to what $M$ signifies in the problem definition, which should be clarified in sections 3 and 4.2.
>
> **AQ3:** Thanks for your kind reminder. $M$ indicates a sub-region $\mathbf{I}^{M}$ formed by masking part of image $\mathbf{I}$. We have clarified this in the revised manuscript.
>
> ----
>
> **Q4:** Lemmas 1 and 2 alongside equations 10 and 11 are already discussed in the problem definition in section 3, thus should be removed with referencing.
>
> **AQ4:** Thanks for your kind reminder. Lemma 1 and 2 serve to illustrate that the function $\mathcal{F}$ we designed satisfies the definition of submodular function in section 3, with corresponding proofs provided. We have removed Equations 10 and 11 to avoid duplication.
>
> ----
>
> **Q5:** Theorem 1 is prior art (Nemhauser et al., 1978) and should just be cited.
>
> **AQ5:** Thanks for your kind reminder, we have cited this reference next to Theorem 1.
>
> [Reference H] George L Nemhauser, Laurence A Wolsey, and Marshall L Fisher. An analysis of approximations for maximizing submodular set functions—i. Mathematical programming, 14:265–294, 1978.
>
> ----
>
> **Q6:** Although optional, it would be good to have a set of notations as this paper encapsulates multiple domains in Machine Learning.
>
> **AQ6:** Thanks for you helpful suggestion. Due to the length of the paper, we have added the notations in Appendix Section A and Table 5 in the revised manuscript:
>
> Table 5: Some important notations used in this paper.
>
> | Notation                      | Description                                                  |
> | ----------------------------- | ------------------------------------------------------------ |
> | $\mathbf{I}$                  | an input image                                               |
> | $\mathbf{I}^M$                | a sub-region into which $\mathbf{I}$ is divided              |
> | $V$                           | a finite set of divided sub-regions                          |
> | $S$                           | a subset of $V$                                              |
> | $\alpha$                      | an element from $V \setminus S$                              |
> | $k$                           | the size of the $S$                                          |
> | $\mathcal{F}(\cdot)$          | a function that maps a set to a value                        |
> | $\mathcal{A}$                 | saliency map calculated by attribution algorithms            |
> | $N \times N$                  | the number of divided patches of $\mathbf{I}$                |
> | $m$                           | the number of sub-regions into which the image $\mathbf{I}$ is divided |
> | $d$                           | the number of patches in $\mathbf{I}^M$                      |
> | $\mathbf{x}$                  | an input sample                                              |
> | $\mathbf{y}$                  | the one-hot label                                            |
> | $u$                           | the predictive uncertainty, and the value range from 0 to 1  |
> | $F_{u}(\cdot)$                | a deep evidential network for calculating the $u$ of $\mathbf{x}$ |
> | $F(\cdot)$                    | a traditional network encoder                                |
> | $K$                           | the number of classes                                        |
> | $\mathrm{dist}(\cdot, \cdot)$ | a function to calculate the distance between two feature vectors |
> | $\boldsymbol{f}_{s}$          | the target semantic feature vector                           |
>
> ----
>
> **Q7:** Experiments in section 5 indicate that the proposed method demonstrates significant improvements over existing methods. This shows the generalizability of the approach irrespective of the underlying model, which I believe can be highlighted for better impact.
>
> **AQ7:** Thanks for your helpful suggestion. We highlight this advantage in Section 5.2.

---

> > ### Comment · Reviewer_kDMz · 2023-11-22
> >
> > I thank the authors for your response. The detailed response to the concerns raised during the review process is highly appreciated. The updated paper (after incorporation of all changes) is definitely better readable and clearly points out the novelty of the approach. I have thus decided to increase my score to 8.

---

### Author Response · Authors · 2023-11-20
**General Response: Revision Summary**

We extend our sincere gratitude to all reviewers for their valuable and constructive feedback, which has significantly enhanced the quality of our paper. In response to their suggestions, we have undertaken additional experiments and made several revisions, all of which are highlighted in ***blue*** in the PDF file. Below is a summary of these revisions:

- Added a list of notations in Appendix Section A.
- Updated Figure 1 to add the Input Image $\mathbf{I}$ and the attribution map $\mathcal{A}$.
- Updated Figure 2 to label the Input Image as $\mathbf{I}$ and the attribution map as $\mathcal{A}$.
- Revised Section 4.1 on Sub-region Division for enhanced clarity.
- Included detailed explanations for each score in Section 4.2.
- Detailed the steps of sub-region division in Algorithm 1.
- Removed duplicate equations from Lemma 1 and Lemma 2.
- Reported the attribution ability of our method for incorrectly predicted samples under different network backbones in Appendix D.
- Conducted an ablation study on the effect of sub-region division size $N \times N$ in incorrect sample attribution, detailed in Appendix F.
- Discussed additional limitations of our study in Appendix J.
- Cited several important references previously omitted in the original manuscript.
- Corrected various unclear expressions and grammatical errors.

---

### Meta-Review · Area_Chair_6UD2 · 2023-12-11

**Metareview:**

This is a strong paper. The paper presents a novel combinatorial approach for image attribution problem and the experimental results are promising! The reviewers have all voted positively to accept this paper, and I agree with their assessments. The authors have also addressed almost all concerns of the reviewers. I would encourage the authors to consider updating the paper with the reviewer feedback.

**Justification For Why Not Higher Score:**

N/A

**Justification For Why Not Lower Score:**

As mentioned above, this is a comprehensive paper that provides a combinatorial approach to the image attribution problem. The authors have addressed most concerns of the reviewers and this paper would be an excellent addition to ICLR 2024!

---

### Decision · Program_Chairs · 2024-01-16

Accept (oral)